# GNPNAT1 Serves as a Prognostic Biomarker Correlated with Immune Infiltration and Promotes Cancer Cell Metastasis through Stabilization of Snai2 in Lung Adenocarcinoma

**DOI:** 10.3390/biomedicines12071477

**Published:** 2024-07-04

**Authors:** Jinqi He, Faxiang Li, Zihan Jing, Xingmei Ren, Dexin Jia, Yuan Zeng, Yan Yu

**Affiliations:** 1Department of Medical Oncology, Harbin Medical University Cancer Hospital, Harbin 150040, China; hejinqi2014@163.com (J.H.); jingzihan8023@163.com (Z.J.); renxiaobai15@163.com (X.R.); xinxinxiao5@126.com (D.J.); 15172433915@163.com (Y.Z.); 2Department of Medical Oncology, The Central Hospital of Shaoyang, Shaoyang 422000, China; lfxsyzxyy@126.com

**Keywords:** LUAD, *GNPNAT1*, *Snai2*, proliferation, metastasis, prognosis

## Abstract

Background: Lung cancer is a common malignant tumor with high morbidity and mortality rate. Glucosamine 6-phosphate N-acetyltransferase (*GNPNAT1*), which serves as a critical enzyme in hexosamine biosynthetic pathway (HBP), has been identified as a metastasis-associated gene and is upregulated in lung adenocarcinoma (LUAD). However, the exact role and related mechanism of *GNPNAT1* in LUAD metastasis remain unknown. Methods: We analyzed the expression of *GNPNAT1* in the public databases and confirmed the results by immunohistochemistry (IHC). The biological functions of *GNPNAT1* in LUAD were investigated based on The Cancer Genome Atlas (TCGA). Correlations between *GNPNAT1* and cancer immune characteristics were analyzed via the Estimation of Stromal and Immune cells in Malignant Tumor tissues using Expression data (ESTIMATE) and Cell-type Identification by Estimating Relative Subsets of RNA Transcript (CIBERSORT) R package. The underlying mechanisms of altered *GNPNAT1* expression on LUAD cell tumorigenesis, proliferation, migration, invasion, and metastasis were explored in vitro and in vivo. Results: We demonstrated that *GNPNAT1* expression was significantly increased in LUAD and negatively associated with the overall survival (OS) of patients. *hsa-miR-1-3p* and *hsa-miR-26a-5p* were identified as upstream miRNA targets of *GNPNAT1*. *GNPNAT1* was associated with the infiltration levels of CD8 T cells, memory-activated CD4 T cells, NK cells resting, macrophages M0, macrophages M1, neutrophils, gamma delta T cells, and eosinophils, while it was negatively correlated with memory-resting CD4 T cells, regulatory T cells (Tregs), resting NK cells, monocytes, resting dendritic cells, and resting mast cells. *GNPNAT1* knockdown significantly inhibited proliferation, migration, invasion, epithelial–mesenchymal transition (EMT) process, and metastasis of LUAD cells, while overexpression of *GNPNAT1* revealed the opposite effects. Rescue assay showed that *Snai2* knockdown reversed *GNPNAT1*-induced LUAD cells migration, invasion, and EMT. Mechanistically, *GNPNAT1* promoted cancer cell metastasis via repressing ubiquitination degradation of *Snai2* in LUAD. Conclusions: Taken together, these data indicate that *GNPNAT1* serves as a prognostic biomarker for LUAD patient. Additionally, *GNPNAT1* is critical for promoting tumorigenesis and metastasis of LUAD cells and may be a potential therapeutic target for preventing LUAD metastasis.

## 1. Introduction

Lung cancer is one of the most common malignant tumors and the primary cause of cancer death worldwide [1,2]. Lung adenocarcinoma (LUAD) is the most common pathologic type of lung cancer, comprising over 50% of all lung cancer cases and almost 40% of death related to lung cancer [3,4,5]. Over the past decade, significant improvements have been achieved in early diagnosis and clinical treatment of LUAD. However, the overall survival (OS) of LUAD patients remains unsatisfactory. The 5-year survival rate of advanced or metastatic LUAD patients remains below 15% and has not increased in the past few years due to its usually invasive and metastatic activity [6,7,8]. Metastasis is the leading cause of death for LUAD patients. Although many studies have been conducted on LUAD metastasis, our understanding of the molecular mechanism of LUAD metastasis remains limited and unclear. Therefore, identifying new biomarkers to improve the OS of LUAD patients and overcome LUAD metastasis is of great significance.

Glucosamine-6-phosphate N-acetyltransferase 1 *(GNPNAT1)*, a small dimeric protein located in the Golgi matrix and endomembrane, serves as the rate-limiting enzyme in the second step of the hexosamine biosynthetic pathway (HBP). It catalyzes the conversion of glucosamine-6-phosphate to N-acetylglucosamine-6-phosphate [9,10]. The function of GNPNAT1 is depicted in detail in Appendix A. Emerging evidence suggests aberrant *GNPNAT1* expression in various cancers, including castrated-resistant prostate cancer (CRPC), breast cancer (BC) and LUAD [11,12,13,14,15,16,17]. In CRPC, *GNPNAT1* downregulation is observed in CRPC tissues and contributes to metastasis by regulating PI3K/AKT signal pathway [11]. Studies on breast cancer have demonstrated a positive correlation between *GNPNAT1* upregulation and poor patient prognosis, along with a negative correlation with immune infiltration [16,17]. Notably, *GNPNAT1* deletion has been shown to reduce breast cancer cell proliferation and invasion. Similarly, several studies have reported a positive correlation between *GNPNAT1* upregulation and poor prognosis in LUAD patients [12,13,14,15]. Additionally, *GNPNAT1* has been implicated in promoting the growth, proliferation, migration, and invasion of LUAD cells [18,19,20]. However, the precise regulatory mechanisms by which *GNPNAT1* influences LUAD metastasis remain to be elucidated.

To investigate the potential of *GNPNAT1* as a prognostic biomarker for LUAD, this study evaluated its expression in LUAD patients and its impact on their prognosis. Additionally, the correlation between *GNPNAT1* expression and immune infiltration within the tumor microenvironment was explored. Subsequently, in vitro and in vivo experiments were conducted to elucidate the role of *GNPNAT1* in the proliferation, migration, invasion, and epithelial-to-mesenchymal transition (EMT) process of LUAD cells and in nude mice models. Our findings demonstrated that *GNPNAT1* knockdown could suppress tumorigenesis and metastasis of LUAD cells. Mechanistically, we found that *GNPNAT1* could promote cancer cell metastasis by stabilizing Snai2 in LUAD. Overall, our results suggest that *GNPNAT1* represents a novel prognostic biomarker and a promising therapeutic target for LUAD patients.

## 2. Materials and Methods

### 2.1. Data Collection

The RNA-sequencing (RNA-seq) data and corresponding clinical data for LUAD patients were obtained from The Cancer Genome Atlas (TCGA-LUAD cohort; https://portal.gdc.cancer.gov/, accessed on 15 October 2022). Additionally, six independent datasets (GSE32863 [21], GSE40791 [22], GSE75037 [23], GSE115002 [24], GSE13213 [25], and GSE72094 [26]) were downloaded from the Gene Expression Omnibus (GEO; https://www.ncbi.nlm.nih.gov/, accessed on 9 January 2023)). The expression level of *GNPNAT1* in LUAD tissues was analyzed using multiple resources: The Tumor Immune Estimation Resource (TIMER2; http://timer.comp-genomics.org, accessed on 6 January 2023), Gene Expression Profiling Interactive Analysis 2 (GEPIA2; http://gepia2.cancer-pku.cn/, accessed on 6 January 2023), and the following GEO datasets: GSE32863 [21], GSE40791 [22], GSE75037 [23] and GSE115002 [24]. Furthermore, survival analyses for patients with high and low GNPNAT1 expression were performed using data from the TCGA-LUAD cohort, GSE13213 [25], and GSE72094 [26]. The clinicopathological characteristics of the different datasets are presented in Appendix A.

### 2.2. Immune Infiltration of GNPNAT1 in LUAD

The R package “Estimation of Stromal and Immune cells in Malignant Tumor tissues using Expression” (ESTIMATE) (https://sourceforge.net/projects/estimateproject/, accessed on 20 December 2023) was employed to evaluate the levels of immune cell infiltration, stromal content, and the stromal-immune comprehensive score for each LUAD sample within the TCGA-LUAD cohort [27]. Additionally, the R package “Cell-type Identification by Estimating Relative Subsets of RNA Transcript” (CIBERSORT) (https://cibersortx.stanford.edu/, accessed on 20 December 2023) was utilized to assess the infiltration status of 22 distinct immune cell types [28]. Finally, Spearman’s correlation analysis was performed to identify potential correlations between immune cell infiltration and GNPNAT1 expression.

### 2.3. Cell Lines and Cell Culture

The human LUAD cell lines PC9, H1299, A549, H1993, HCC827, and H1650 were obtained from the Heilongjiang Cancer Institute (https://www.hrbmu.edu.cn/ssky/info/1025/1125.htm, accessed on 20 December 2023, Harbin, China). H1299, A549, H1993, HCC827, and H1650 cells were cultured in Roswell Park Memorial Institute (RPMI)-1640 medium (Gibco^®^, Grand Island, NY, USA) supplemented with 10% fetal bovine serum (FBS; PAN, Biotech GmbH, Marktredwitz, Germany). PC9 cells were maintained in Dulbecco’s Modified Eagle’s Medium (DMEM; Gibco^®^) supplemented with 10% FBS. All cell lines were incubated at 37 °C in a humidified incubator with 5% CO_2_.

### 2.4. Cell Transfection

To manipulate *GNPNAT1* expression, this study employed transient knockdown and overexpression strategies. For knockdown, small interfering RNAs (siRNAs) targeting *GNPNAT1* (si-*GNPNAT1*#1, #2, and #3) from RiboBio (Guangzhou, China) were transfected into PC9 and H1299 cells using jetPRIME reagent (Polyplus, Strasbourg, France). Conversely, *GNPNAT1* overexpression was achieved by transfecting H1993 cells with pCMV3-GNPNAT1 (Shen Gong, Shanghai, China) via jetPRIME. All transfections included negative controls—either non-targeting siRNAs or empty pCMV3 vectors. To investigate the role of Snai2 in *GNPNAT1*-mediated effects, si-Snai2#1 and #2 were used to transiently knockdown Snai2 in H1993 cells overexpressing GNPNAT1. All procedures strictly followed the manufacturer’s protocols. The siRNA sequences are provided in Appendix A. Based on knockdown efficiency, si-*GNPNAT1*#3 was chosen to generate lentiviruses for stable *GNPNAT1* knockdown in H1299 cells, which were subsequently selected using puromycin.

### 2.5. Quantitative Real-Time PCR (qRT-PCR)

RNA isolation, cDNA synthesis, and qRT-PCR analysis: Total RNA was isolated from cells using TRK lysis buffer (R6834-01, Omega Bio-Tek, Norcross, GA, USA) following the manufacturer’s protocol. cDNA was synthesized using 5xFastKing-RT SuperMix (KRT11802, Tiangen, Beijing, China). Quantitative real-time PCR (qRT-PCR) analysis was performed using FastStart Universal SYBR Green Master (4913850001, ROX, Shanghai, China). The relative expression level of target mRNA was calculated using the 2^−ΔΔCt^ method and normalized to *GAPDH* as an internal control. Primer sequences used for qRT-PCR are listed in Appendix A.

### 2.6. Western Blot Assay

For protein extraction, cells were harvested, lysed in radioimmunoprecipitation assay (RIPA) buffer (P0013E-1, Beyotime, Beijing, China) containing phenylmethylsulfonyl fluoride (PMSF), and protein concentration was determined using the bicinchoninic acid (BCA) protein assay kit (P0010S, Beyotime, China). Equal amounts of protein (30 μg) were loaded onto 10% or 12.5% sodium dodecyl sulfate–polyacrylamide gel (SDS-PAGE) for electrophoresis, followed by transfer onto polyvinylidene difluoride (PVDF) membranes via electroblotting. The membranes were blocked with 5% non-fat dry milk in phosphate-buffered saline with 0.1% Tween-20 (PBST) for 1 h at room temperature. Subsequently, membranes were incubated overnight at 4 °C with specific primary antibodies, followed by horseradish peroxidase (HRP)-conjugated secondary antibodies for 1 h at room temperature. The Chemistar ^TM^ High-sig ECL Western Blotting Substrate (180-5001, Tanon, Shanghai, China) was used to visualize the specific protein bands on the membrane. The antibodies against proteins are listed in Appendix A.

### 2.7. Co-Immunoprecipitation Assay

PC9 cells transfected with either si-NC or si-*GNPNAT1*#3 were lysed in RIPA lysis buffer (89900, Thermo Fisher Scientific, Waltham, MA, USA) to obtain protein supernatants. The supernatants were incubated with the corresponding antibodies overnight at 4 °C, followed by incubation with protein G magnetic beads for an additional 2–4 h at 4 °C. The protein G magnetic beads were then separated using a magnetic rack and washed with PBST. Finally, the immunoprecipitated proteins were used for Western Blot analysis.

### 2.8. Wound-Healing Assay

H1299, PC9, and H1993 cells were seeded onto 6-well plates containing a complete culture medium. Once the cells reached 85–90% confluence, a scratch wound was created using a sterile micropipette tip. The wound gap was then visualized and documented using microscopic examination.

### 2.9. Transwell Assay

Transwell chambers (Corning Incorporated, Corning, NY, USA) were employed to assess tumor cell migration and invasion. For the migration assay, 2 × 10^4^ tumor cells suspended in 200 μL of serum-free medium (medium without 10% FBS) were seeded in the upper chambers. The lower chambers were filled with 600 μL of medium supplemented with 10% FBS to serve as a chemoattractant. For the invasion assay, 2 × 10^4^ tumor cells suspended in 200 μL of serum-free medium were seeded in the upper chambers containing Matrigel-coated inserts. Following incubation, migrated or invaded tumor cells were fixed with 4% paraformaldehyde, stained with 0.1% crystal violet, and subsequently imaged and quantified using a light microscope.

### 2.10. Cell Proliferation Assay

The Cell Counting Kit-8 (CCK-8) assay (CK04-1000t, Dojindo, Kumamoto, Japan) was used to assess cell viability. H1993 cells transfected with either pCMV3 empty vector or pCMV3-*GNPNAT1* plasmid, and PC9 and H1299 cells transfected with either a non-targeting control siRNA (si-NC) or three different *GNPNAT1*-targeting siRNAs (si-*GNPNAT1*#1, si-*GNPNAT1*#2, and si-*GNPNAT1*#3) were seeded into 96-well plates at a density of 3 × 10^3^ cells per well. Following incubation for 24, 48, and 72 h at 37 °C, 10 μL of CCK-8 solution was added to each well. The absorbance at 450 nm was then measured using a microplate reader. The EdU incorporation assay was performed using the EdU Apollo^®^ 488 In Vitro Kit (C10310-3, RiboBio, Guangzhou, China) according to the manufacturer’s instructions. Briefly, cells were treated and incubated as described above. EdU staining was performed, and fluorescence microscopy was used to capture images of EdU incorporation.

### 2.11. Immunohistochemistry (IHC) Staining Assay

For IHC analysis, tissue microarrays (HLungA150CS03, Shanghai Outdo Biotech, Shanghai, China) containing 75 matched pairs of primary lung adenocarcinoma samples and adjacent normal tissues were obtained commercially. Mounted tissue sections were deparaffinized in xylene, dehydrated through a graded ethanol series, and rehydrated in water. Endogenous peroxidase activity was quenched with 3% hydrogen peroxide. Antigen retrieval was performed using 0.01 M citrate buffer (pH 6.0). To block non-specific antibody binding, sections were incubated with normal goat serum. Subsequently, sections were incubated overnight at 4 °C with a 1:500 dilution of a primary antibody against *GNPNAT1*, followed by incubation with horseradish peroxidase (HRP)-conjugated goat anti-rabbit IgG secondary antibody for 1 h at room temperature. Immunoreactivity was visualized using 3,3′-diaminobenzidine (DAB) chromogenic substrate, followed by counterstaining with hematoxylin and eosin. Finally, sections were dehydrated, mounted, and examined under a light microscope. Two experienced pathologists independently evaluated the IHC staining intensity and the proportion of positive tumor cells according to established scoring criteria [29,30]. The IHC staining intensity was scored on a four-point scale: 0 (negative), 1 (weakly positive), 2 (moderately positive), and 3 (strongly positive). The proportion of positive tumor cells was also scored: 0 (0%), 1 (1–25%), 2 (26–50%), 3 (51–75%), and 4 (76–100%). The final IHC score was calculated by summing the staining intensity score and the proportion score. Based on the median of these combined scores, LUAD patients were categorized into high- and low-*GNPNAT1*-expression groups.

### 2.12. Ubiquitination Assay

PC9 cells were transfected with either control siRNA (si-NC) or the indicated GNPNAT1-targeting siRNAs. Following 48 h of culture, MG132 was added to the cells to inhibit Snai2 protein degradation. Subsequently, cell lysates were prepared using denaturing lysis buffer. Finally, the denatured protein lysates were used for Western blotting (WB) and immunoprecipitation (IP) analyses.

### 2.13. Cycloheximide (CHX) Pulse-Chase Assay

PC9 cells were seeded into 12-well plates at a density of 2 × 10^5^ cells per well and transfected with the indicated siRNAs or a non-targeting control siRNA. After 48 h of incubation, the supernatant was collected from the PC9 cell cultures, denatured, and subjected to WB analysis.

### 2.14. In Vivo Assay

All animal procedures were conducted in accordance with the principles outlined in the Declaration of Helsinki. The study protocol was reviewed and approved by the ethics committee of Harbin Medical University Cancer Hospital (approval number: KY2023-26). Four- to six-week-old female BALB/c nude mice were purchased from Charles River Co., Ltd. (Beijing, China). Mice were housed in a temperature-controlled environment with a 12 h light/dark cycle and provided with ad libitum access to standard laboratory rodent chow and water. For the in vivo xenograft assays, 3 × 10^6^ H1299 cells stably transfected with sh-*GNPNAT1* (Miaolingbio, Wuhan, China) or a control vector were separately injected subcutaneously into the right subaxillary region of nude mice (*n* = 4 per group). Tumor volume (V) was measured using calipers at regular intervals and calculated according to the formula: V = (length × width^2^)/2. After four weeks of subcutaneous inoculation, mice were anesthetized using an approved protocol and then euthanized. Tumors were harvested and fixed in 4% paraformaldehyde solution. For the in vivo metastasis assays, 3 × 10^6^ H1299 cells stably transfected with sh-*GNPNAT1* or a control vector were injected into the tail vein of nude mice (*n* = 4 per group). After seven weeks of tail vein injection, mice were anesthetized using an approved protocol and then euthanized. Lung tissues were collected and fixed in 4% paraformaldehyde solution.

### 2.15. Statistical Analysis

All experiments were performed in triplicate. Statistical analyses were conducted using R software (version 4.2.2) and GraphPad Prism 9. Data were presented as mean ± SEM. Kaplan–Meier analysis with a log-rank test was employed for survival analysis. ROC analysis assessed the predictive accuracy of the 4-immunomodulatory gene signature in LUAD patients. To determine if *GNPNAT1* and the 4-immunomodulatory gene signature were independent predictors of OS in LUAD patients, univariate and multivariate Cox proportional hazard models were performed. The rms package generated a general linear model for predicting OS in LUAD patients. Spearman’s rank correlation analysis evaluated the correlation between *GNPNAT1* and corresponding miRNAs. Student’s *t*-test assessed differences between the two groups, with *p*-values < 0.05 considered statistically significant.

## 3. Results

### 3.1. GNPNAT1 was Highly Expressed in LUAD

Bioinformatics analysis was performed to investigate *GNPNAT1* expression in LUAD. Figure 1A,B demonstrate high *GNPNAT1* expression documented in various malignant tumors, including LUAD and LUSC. Additionally, an analysis of multiple datasets (GSE32863 [21], GSE40791 [22], GSE75037 [23] and GSE115002 [24]) revealed significantly higher *GNPNAT1* expression in LUAD compared to normal tissues (Figure 1C–F). Immunohistochemistry further confirmed this finding, showing significantly stronger GNPNAT1 expression in LUAD tissues than in normal controls (Figure 1G,H).

### 3.2. GNPNAT1 Was Associated with Poor Prognosis in LUAD

Analysis of TCGA-LUAD cohort and gene expression datasets GSE72094 [26] and GSE13213 [25] revealed a significant correlation between high GNPNAT1 expression and poor OS in LUAD patients (Figure 2A–C). The Wilcoxon test further demonstrated a positive correlation between GNPNAT1 expression and clinicopathological characteristics, including TNM stage, tumor stage, and lymph node metastasis stage (Figure 2D). Univariate and multivariate Cox proportional hazard models confirmed *GNPNAT1* as an independent prognostic factor for LUAD patient OS (Figure 3A,B). Additionally, a nomogram incorporating *GNPNAT1* expression and clinical variables (age, gender, TNM stage, T stage, N stage, M stage) was constructed to directly estimate the 1-, 3-, and 5-year OS for LUAD patients (Figure 3C,D).

### 3.3. Has-miR-1-3p and Has-miR-26a-5p Were the miRNA Targets of GNPNAT1

To further explore the potential biological functions of *GNPNAT1* in LUAD, we identified differentially co-expressed genes in the TCGA-LUAD cohort. We selected genes with a correlation coefficient (r) greater than 0.55 and a *p*-value less than 0.001 compared to GNPNAT1 expression. This analysis revealed 288 positively correlated genes and 56 negatively correlated genes. The top 11 positively and 10 negatively correlated genes are listed in Figure 4A.

Subsequently, we performed Gene Ontology (GO) and Kyoto Encyclopedia of Genes and Genomes (KEGG) pathway enrichment analyses using all 344 co-expressed genes. The GO analysis results indicated that these genes were primarily enriched in functions related to cell division, including mitotic nuclear division, nuclear division, chromosome segregation, chromosomal region, condensed chromosome, and ATP hydrolysis activity (Figure 4B). Similarly, the KEGG analysis revealed significant enrichment in pathways associated with cell cycle progression, oocyte meiosis, and DNA replication (Figure 4C). To identify potential upstream miRNA regulators of GNPNAT1, we retrieved candidate miRNAs predicted to bind to *GNPNAT1* from the ENCORI database (https://rnasysu.com/encori/index.php, accessed on 30 November 2023). As shown in Figure 4D, 51 miRNAs were identified as potential regulators. Correlation analysis between these miRNAs and GNPNAT1 expression revealed that *hsa*-*miR*-*1*-*3p*, *hsa*-*miR*-*26a*-*5p*, *hsa*-*miR*-*664b*-*3p*, *hsa*-*miR*-*135b*-*5p*, and *hsa*-*miR*-*27b*-*3p* exhibited significant negative correlations (Figure 4E–I). A differential expression analysis of these miRNAs in LUAD tissues compared to normal tissues was then performed. We found that *hsa*-*miR*-*664b*-*3p*, *hsa*-*miR*-*135b*-*5p*, and *hsa*-*miR*-*27b*-*3p* expression was significantly higher in LUAD tissues (Figure 5C–E), while *hsa*-*miR*-*1*-*3p* and *hsa*-*miR*-*26a*-*5p* expression was lower (Figure 5A,B). Finally, survival analysis was conducted to assess the prognostic value of these miRNAs. The OS of LUAD patients was significantly positively correlated with the expression of hsa-miR-1-3p, *hsa-miR-26a-5p*, and *hsa-miR-135b-5p* (Figure 5F,G,I). Conversely, no significant correlation between OS and the expression of hsa-miR-664b-3p or hsa-miR-27b-3p was observed (Figure 5H,J). Based on the combined results of differential expression, survival analysis, and their negative correlation with GNPNAT1, we propose that hsa-miR-1-3p and hsa-miR-26a-5p are the most likely upstream miRNA targets regulating *GNPNAT1* expression in LUAD.

### 3.4. Correlation of GNPNAT1 Expression with Immune Infiltration in LUAD

ESTIMATE analysis suggested a higher abundance of immune cells in the low-*GNPNAT1*-expression group compared to the high-expression group (Figure 6A). CIBERSORT analysis revealed increased infiltration levels of CD8+ T cells, memory-activated CD4+ T cells, resting NK cells, M0 macrophages, M1 macrophages, and neutrophils in the high-*GNPNAT1*-expression group (Figure 6B). Conversely, infiltration levels of memory-resting CD4+ T cells, regulatory T cells (Tregs), monocytes, resting dendritic cells, and resting mast cells were decreased in the high-*GNPNAT1*-expression group (Figure 6B). Further analysis demonstrated a positive correlation between *GNPNAT1* expression and the infiltration levels of CD8+ T cells, memory-activated CD4+ T cells, resting NK cells, M0 macrophages, M1 macrophages, neutrophils, gamma delta T cells, and eosinophils. Conversely, a negative correlation was observed between GNPNAT1 expression and the infiltration levels of memory-resting CD4+ T cells, Tregs, resting NK cells, monocytes, resting dendritic cells, and resting mast cells (Figure 6C).

### 3.5. Construction of a Prognostic Signature Based on GNPNAT1-Associated Immunomodulator Genes

Immunomodulatory genes play a crucial role in regulating the functions of the immune system. We retrieved 69 immunomodulatory genes from the TISIDB database (http://cis.hku.hk/TISIDB/, accessed on 30 November 2023). The correlation between *GNPNAT1* expression and these immunomodulatory genes is presented in Figure 6D. Univariate Cox proportional hazards model analysis identified nine GNPNAT1-associated immunomodulatory genes significantly correlated with OS of LUAD patients (Figure 7A). Subsequently, a prognostic signature incorporating *CD276*, *CD40LG*, *PVR*, and *REAT1E* was constructed using a multivariate Cox proportional hazards model (Figure 7B). The median risk score was used as the cutoff to categorize patients from the TCGA-LUAD cohort into high-risk and low-risk groups. Figure 7C displays the risk score, survival status, and gene expression heatmap of these prognostic immunomodulatory genes. Patients with high-risk scores exhibited significantly poorer OS compared to those with low-risk scores (Figure 7D). The area under the curve (AUC) of the receiver operating characteristic (ROC) curve for the risk score was 0.688. Combining the risk score with the clinical stage further improved the AUC to 0.743, exceeding the predictive power of other individual clinical variables (Figure 7E). Both univariate and multivariate Cox regression analyses confirmed that the risk score served as an independent prognostic factor for OS in LUAD patients (Figure 8A,B). Nomogram analysis and calibration curves further validated the reliability of the 4-gene signature in predicting OS for LUAD patients (Figure 8C–F).

### 3.6. GNPNAT1 Promoted LUAD Cell Proliferation

To investigate the potential role of *GNPNAT1* in LUAD cell proliferation, we first evaluated endogenous *GNPNAT1* expression levels in six LUAD cell lines: H1650, HCC827, PC9, H1993, H1299, and A549. The results demonstrated that PC9 and H1299 cells exhibited higher *GNPNAT1* expression compared to A549, H1650, HCC827, and H1993 cells, with the lowest expression observed in H1993 cells (Figure 9A). Based on these findings, H1299, PC9, and H1993 cells were selected for subsequent functional experiments. Western blotting and qRT-PCR analyses confirmed successful overexpression or knockdown of GNPNAT1 in H1993 (Figure 9F,G) and H1299 (Figure 9B,C) and PC9 cells (Figure 9D,E) at both the protein and transcriptional levels. Cell proliferation assays, including CCK8, colony formation, and EdU assays, revealed a marked enhancement in the proliferative capacity of LUAD cells overexpressing *GNPNAT1* (Figure 10C,F,I). Conversely, knockdown of *GNPNAT1* significantly reduced the proliferation rate compared to control cells (Figure 10A,D,G; Figure 10B,E,H).

### 3.7. GNPNAT1 Promoted LUAD Cells Migration and Invasion

Cancer cell metastasis depends on the ability of cells to migrate and invade surrounding tissues. To determine whether *GNPNAT1* influences the migration and invasion of LUAD cells, we performed wound healing assays to assess the impact of *GNPNAT1* transfection on H1299, PC9, and H1993 cells. As shown in Figure 11A,B, cells transfected with si-*GNPNAT1* migrated towards the scratch wound at a slower rate compared to control cells in H1299 and PC9 lines. Conversely, H1993 cells with *GNPNAT1* overexpression exhibited enhanced migration (Figure 11C). Furthermore, transwell assays confirmed that GNPNAT1 knockdown significantly decreased the invasive capacity of H1299 (Figure 11D) and PC9 cells (Figure 11E). Conversely, overexpression of *GNPNAT1* in H1993 cells resulted in a marked increase in their invasion ability (Figure 11F). To further investigate the effect of *GNPNAT1* on the proliferative and metastatic potential of LUAD cells in vivo, we conducted animal experiments. As expected, tumors derived from H1299-sh-*GNPNAT1* cells grew slower and were smaller than those from control cells (Figure 12A,C). Additionally, the in vivo metastasis assay revealed a significantly lower number of metastatic lung nodules in mice injected with H1299-sh-*GNPNAT1* cells compared to the control group (Figure 12B)

### 3.8. GNPNAT1 Induced EMT Process of LUAD Cell

Epithelial–mesenchymal transition (EMT) plays a critical and complex role in promoting tumor invasion and metastasis in epithelial-derived carcinomas, including LUAD. Our findings suggested that GNPNAT1 expression levels could influence the EMT process in LUAD cells. H1299 and PC9 cells transfected with si-*GNPNAT1* exhibited a mesenchymal-epithelial transition (MET) phenotype characterized by upregulation of the epithelial marker E-cadherin and downregulation of the mesenchymal markers N-cadherin, Vimentin, and Snai2 (Figure 13A,B). Conversely, H1993 cells overexpressing GNPNAT1 displayed a typical EMT phenotype, evidenced by the upregulation of mesenchymal markers and the transcriptional factor Snai2, alongside a decrease in the epithelial marker E-cadherin (Figure 13C).

### 3.9. Snai2 Was Required in GNPNAT1-Mediated Migration, Invasion, and EMT Processes

*SNAI2*, a well-characterized transcription factor, can induce pathological EMT by suppressing E-cadherin expression. To determine whether Snai2 was essential for *GNPNAT1*’s role in migration, invasion, and EMT, we first analyzed the GEPIA2 database (http://gepia2.cancer-pku.cn/, accessed on 16 September 2022) and found a positive correlation between *GNPNAT1* and Snai2 expression (Figure 14A). Next, the downregulation of Snai2 was observed at the protein level (Figure 14B). Subsequently, a rescue assay showed that Snai2 knockdown dampened *GNPNAT1* overexpression-induced H1993 cell migration and invasion, as evidenced by wound-healing, migration, and invasion assays (Figure 14C,D). In addition, *GNPNAT1*-induced E-cadherin downregulation and N-cadherin upregulation were reversed by Snai2 knockdown in *GNPNAT1*-overexpressing H1993 cells (Figure 14E). Based on the above results, it can be inferred that Snai2 is necessary for *GNPNAT1*-mediated migration, invasion, and EMT processes.

### 3.10. GNPNAT1 Promoted Cancer Cell Metastasis through Stabilization of Snai2

To gain further insights into the potential molecular mechanisms by which *GNPNAT1* regulates LUAD metastasis, we employed Gene Set Enrichment Analysis (GSEA; https://www.gsea-msigdb.org/gsea/login.jsp, accessed on 22 October 2022). The GSEA results revealed a significant enrichment of gene sets associated with cell cycle progression and ubiquitin-mediated proteolysis in LUAD patients with high *GNPNAT1* expression (Figure 15A). Ubiquitination is a crucial post-translational modification that regulates protein stability, trafficking, activity, and protein–protein interactions. Disruptions in the ubiquitin pathway are implicated in various diseases, including cancer. Based on these findings, we hypothesized that *GNPNAT1* promotes LUAD metastasis by stabilizing Snai2. To validate this hypothesis, we treated PC9 cells with CHX or MG132 to inhibit new protein synthesis or prevent protein degradation, respectively. The CHX assay demonstrated that *GNPNAT1* knockdown significantly decreased endogenous *Snai2* protein levels compared to the negative control (si-NC) in PC9 cells (Figure 15B). Conversely, the MG132 assay revealed that endogenous Snai2 protein could be degraded by the ubiquitin–proteasome pathway (Figure 15C). Finally, the ubiquitination assay provided evidence that *GNPNAT1* could suppress the degradation of endogenous *Snai2* by inhibiting the ubiquitin–proteasome pathway (Figure 15D).

## 4. Discussion

Limited effective therapeutic targets for controlling cancer metastasis and mortality remain a significant challenge. Our study identified *GNPNAT1* as a prognostic marker for LUAD patients and a potential target for LUAD-specific molecular therapies. We demonstrated that *GNPNAT1* could promote tumorigenesis and metastasis of LUAD cells.

Analysis of multiple databases, including TIMER 2.0 (http://timer.comp-genomics.org, accessed on 30 November 2023), GEPIA2 (http://gepia2.cancer-pku.cn/, accessed on 30 November 2023), and the datasets GSE32863, GSE40791, GSE75037, and GSE115002, revealed significantly higher *GNPNAT1* expression in LUAD patients compared to controls. Tissue microarray analysis further validated this finding, demonstrating elevated GNPNAT1 expression in LUAD tissues compared to normal tissues. Furthermore, an analysis of the TCGA-LUAD cohort, alongside datasets GSE72094 and GSE13213, indicated that high GNPNAT1 expression correlated with a poorer OS for LUAD patients. Correlation analysis revealed a positive association between *GNPNAT1* expression and clinicopathological characteristics, including tumor stage, lymph node metastasis, and clinical stage (TNM stage). Both univariate and multivariate Cox proportional hazard models identified *GNPNAT1* as an independent prognostic factor for OS in LUAD patients. Additionally, nomogram analysis demonstrated the accuracy of *GNPNAT1* in predicting LUAD OS, with 1-, 3-, and 5-year OS rates of 91.5%, 71.6%, and 46.2%, respectively. Collectively, these findings strongly suggest that *GNPNAT1* functions as an oncogene in LUAD and is closely linked to poor prognosis and disease progression.

A growing body of evidence shows that miRNAs are a class of small, non-coding RNAs known to regulate various physiological and pathological processes, including cell proliferation, migration, apoptosis, invasion, and angiogenesis, by targeting specific genes [31,32]. In this study, we identified *hsa*-*miR*-*1*-*3p* and *hsa*-*miR*-*26a*-*5p* as upstream miRNA targets of *GNPNAT1* and demonstrated a negative correlation between their expression levels and *GNPNAT1* expression in LUAD tissues. Furthermore, our findings revealed that *GNPNAT1* promotes key hallmarks of LUAD progression, including cell proliferation, migration, invasion, EMT, and metastasis. Based on these observations, we hypothesize that *hsa*-*miR*-*1*-*3p* and *hsa*-*miR*-*26a*-*5p* may act as tumor suppressors in LUAD by inhibiting *GNPNAT1* expression. Previous studies supported the role of miRNAs in regulating cancer progression. Tan et al. demonstrated that *C1GALT1* promoted migration and proliferation of bladder cancer cells, while its oncogenic potential was suppressed by *miR-1-3p* through direct binding to its 3′-untranslated region (UTR) [33]. Similarly, *miR-1-3p* has been shown to inhibit gastric cancer progression by targeting CENPF [34], and to suppress colorectal cancer cell proliferation and metastasis by targeting *YWHAZ* and regulating EMT [35]. Additionally, Li et al. reported that *miR-1-3p* impeded prostate cancer cell growth and progression by directly targeting *E2F5* and *PFTK1* [36]. In line with our hypothesis regarding the tumor-suppressive role of *miR-26a-5p*, studies have shown that its loss in exosomes promotes tumor cell proliferation, migration, invasion, and metastasis in endometrial cancer [37]. Furthermore, *miR-26a-5p* was found to be downregulated in bladder cancer and to inhibit cancer cell migration and invasion by repressing *PLOD2* [38].

The tumor microenvironment comprises various cellular components, including immune cells, extracellular matrix, fibroblasts, signaling molecules, and surrounding blood vessels. It is critical in tumor growth, metastasis, and patient clinical outcomes. Studies have shown that immune cell infiltration within the tumor can influence a patient’s prognosis. For instance, the tumor-infiltrating lymphocyte grade has been established as an independent factor for predicting sentinel lymph node status in cancer patients [39]. Our findings demonstrated a close correlation between *GNPNAT1* expression and the infiltration of multiple immune cell types, as well as the expression of various immunomodulatory genes. Based on these associations, we developed a 4-gene signature derived from *GNPNAT1*-associated immunomodulatory genes. Survival analysis, ROC analysis, nomogram analysis, and both univariate and multivariate Cox regression analyses all provided strong evidence that this 4-gene signature could reliably predict the OS of LUAD patients. Collectively, these results suggest that *GNPNAT1* plays a crucial regulatory role in the tumor immune microenvironment and the development of LUAD.

Cancer progression is highly dependent on the ability of tumor cells to proliferate, migrate, and invade surrounding tissues. To investigate the functional effects of *GNPNAT1* on these processes in LUAD, we performed a series of assays. The CCK8, EdU, and colony formation assays consistently demonstrated that *GNPNAT1* overexpression significantly promoted the proliferation of LUAD cells. Furthermore, wound-healing and transwell assays revealed that *GNPNAT1* enhanced the migration and invasion of LUAD cells. Finally, in vivo assays further confirmed that *GNPNAT1* promoted the growth and metastasis of LUAD cells. EMT is a well-conserved cellular reprogramming process that plays a pivotal role in cancer metastasis [40]. During EMT, epithelial cells lose their epithelial characteristics and acquire mesenchymal features, gaining the ability to migrate and invade. This process is characterized by the downregulation of epithelial markers (e.g., E-cadherin, ZO-1) and the upregulation of mesenchymal markers (e.g., Vimentin, Fibronectin) and EMT-associated transcription factors (e.g., Snail1, Snai2, Zeb1/2, Twist1/2) [41,42,43,44]. Our findings revealed that *GNPNAT1* could upregulate N-cadherin, Vimentin, and Slug, while downregulating E-cadherin expression in LUAD cells. These observations strongly suggest that *GNPNAT1* can induce the EMT process in LUAD, potentially contributing to its metastatic potential.

EMT is a tightly regulated process controlled by various signaling pathways (e.g., PI3K/AKT and Wnt) and transcription factors. *Snai2*, a transcription factor containing five C-terminal zinc finger domains, is located at chromosome 8q11.21 and encodes a protein of 268 amino acids [45]. *Snai2* overexpression has been documented in numerous cancers, including lung cancer, breast cancer, colorectal cancer, and others [46]. Notably, *Snai2* is considered the most extensively studied EMT-related transcription factor in LUAD [45]. Studies have shown that *Snai2* expression correlates with poor prognosis and may promote cancer cell metastasis in LUAD [45,47,48]. Additionally, *Snai2* protein stability is regulated through the ubiquitin-mediated proteasome pathway [49]. For instance, *lncSNHG15* promotes cancer progression in colon cancer by preventing *Snai2* ubiquitination [50]. Similarly, TNF-α can inhibit *Snai2* ubiquitination in head and neck squamous cell carcinoma by suppressing the NF-κB signaling pathway, promoting EMT and metastasis [51]. Conversely, Dub3 and Pellino have been shown to interact with *Snai2* in breast [51] and lung cancer, respectively, stabilizing the protein and promoting cancer cell migration, invasion, and proliferation [52]. Our study employed rescue experiments to confirm that *Snai2*, a downstream target gene of *GNPNAT1*, plays a critical role in *GNPNAT1*-mediated migration, invasion, and EMT processes in LUAD cells. Furthermore, CHX pulse-chase assays, proteasome inhibition experiments, and ubiquitination assays revealed that *GNPNAT1* regulated the ubiquitin-mediated proteasome pathway in LUAD cells, specifically by suppressing *Snai2* ubiquitination. These findings collectively suggest that *GNPNAT1* promotes cancer cell metastasis in LUAD by stabilizing *Snai2*.

## 5. Conclusions

In conclusion, our findings demonstrate that *GNPNAT1* is highly expressed in LUAD and is associated with immune regulation within the tumor microenvironment. High *GNPNAT1* expression predicted a poor OS for LUAD patients and contributed to tumorigenesis and metastasis of LUAD cells. Mechanistically, we identified *Snai2* stabilization as a key factor by which *GNPNAT1* promotes LUAD cell metastasis. These data collectively suggest that *GNPNAT1* can serve as a prognostic biomarker for LUAD patients and may represent a potential therapeutic target for preventing LUAD metastasis. Future studies will further explore the precise effects of *GNPNAT1* on LUAD growth and metastasis, paving the way for developing novel therapeutic strategies.

## Figures and Tables

**Figure 1 biomedicines-12-01477-f001:**
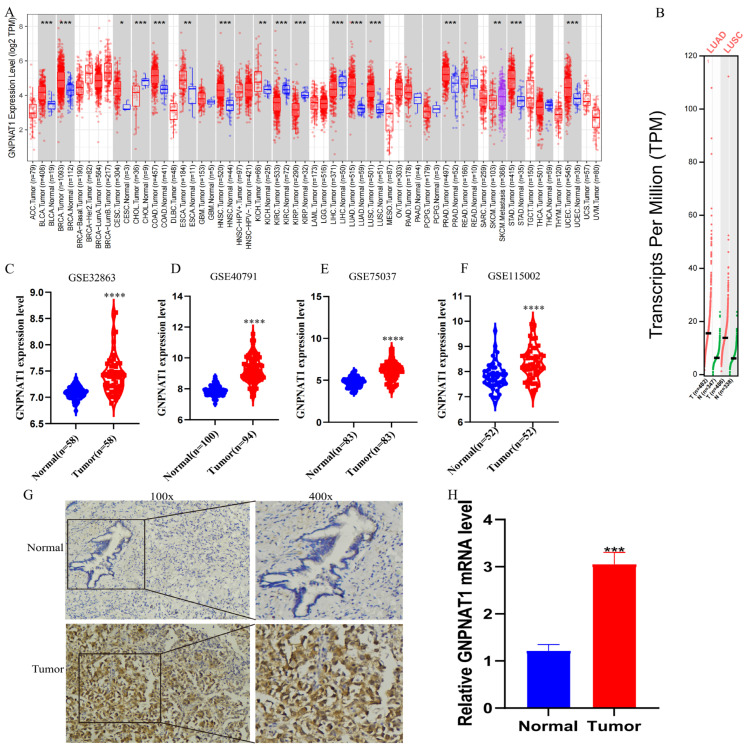
*GNPNAT1* was highly expressed in LUAD. (**A**–**F**) Data from TIMER2, GEPIA2, and GEO database indicated that compared with normal tissues, *GNPNAT1* had a higher expression level in LUAD tissues. * *p* < 0.05, ** *p* < 0.01, *** *p* < 0.001, **** *p* < 0.0001. (**G**) Representative immunohistochemical image of *GNPNAT1* expression in LUAD tissues and matched adjacent noncancerous tissues from tissue microarray (*n* = 75). (**H**) IHC validated that compared with normal tissues, *GNPNAT1* had a higher expression level in LUAD tissues. Data derived from IHC experiment were presented as mean ± SEM, *n* = 3.

**Figure 2 biomedicines-12-01477-f002:**
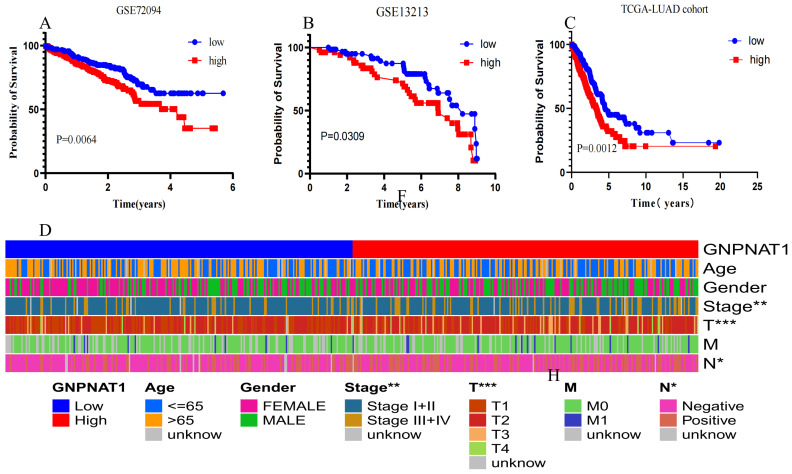
*GNPNAT1* was associated with poor prognosis in LUAD. (**A**–**C**) Data from GEO and TCGA database demonstrated that *GNPNAT1* high expression had a worse OS than *GNPNAT1* low expression using survival analysis. (**D**) Data from TCGA indicated that *GNPNAT1* high expression were positively correlated with T-stage, TNM stage, lymph node metastasis. * *p* < 0.05, ** *p* < 0.01, *** *p* < 0.001.

**Figure 3 biomedicines-12-01477-f003:**
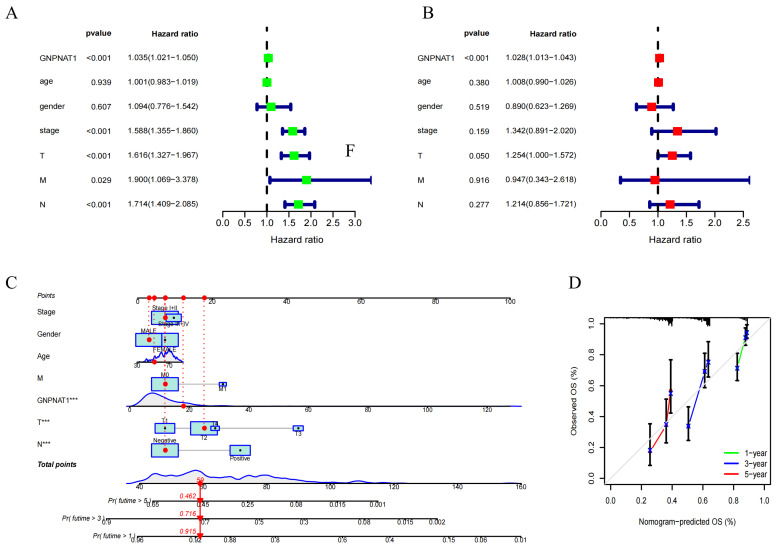
*GNPNAT1* was an independent prognostic factor for LUAD patients. (**A**) The univariate Cox proportional hazard model analysis. (**B**) The multivariate Cox proportional hazard model analysis. (**C**,**D**) Development of a nomogram model on the basis of *GNPNAT1* and calibration of nomogram for *GNPNAT1* at 1, 3, 5 years in TCGA-LUAD cohort. *** *p* < 0.001.

**Figure 4 biomedicines-12-01477-f004:**
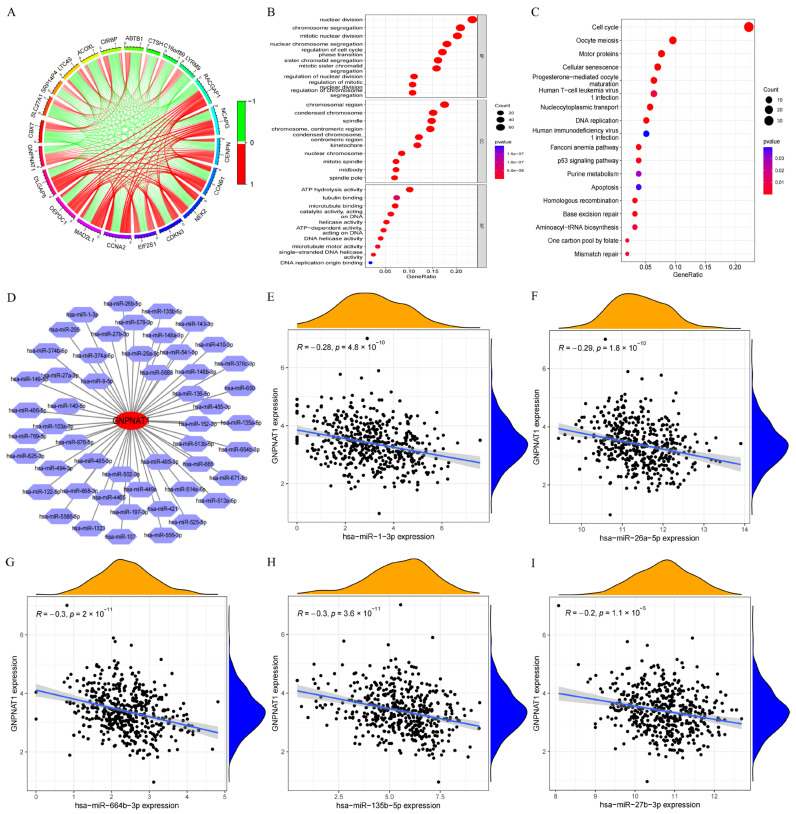
The potential biological functions and miRNA targets of *GNPNAT1* in LUAD. (**A**) The *GNPNAT1*-associated genes in TCGA-LUAD cohort. (**B**,**C**) The Go and KEGG analyses of the *GNPNAT1*-associated genes in TCGA-LUAD cohort. (**D**) The miRNAs interacted with *GNPNAT1* (https://rnasysu.com/encori/index.php, accessed on 30 November 2023). (**E**–**I**) Correlation analysis between *GNPNAT1* and *hsa-miR-1-3p* (**E**), *hsa-miR-26a-5p* (**F**), *hsa-miR-664b-3p* (**G**), *hsa-miR-135b-5p* (**H**), and *hsa-miR-27b-3p* (**I**) in LUAD.

**Figure 5 biomedicines-12-01477-f005:**
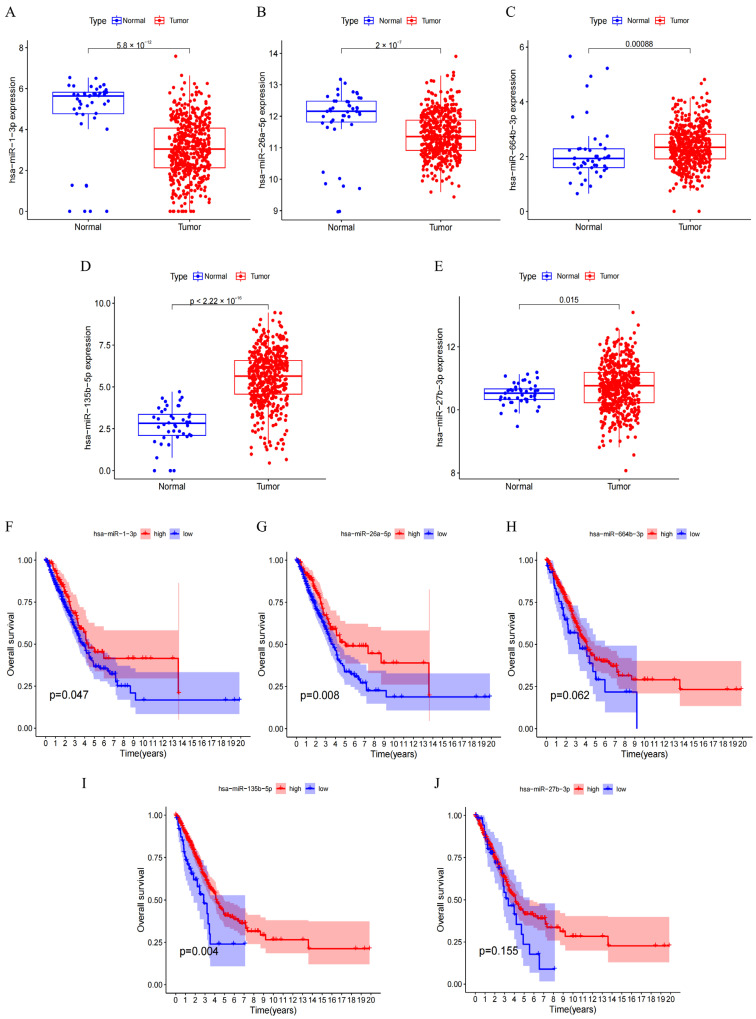
The differential expression and survival analysis for GNPNAT1-associated miRNA targets. (**A**–**E**) Differential expression analysis of *hsa-miR-1-3p* (**A**), *hsa-miR-26a-5p* (**B**), *hsa-miR-664b-3p* (**C**), *hsa-miR-135b-5p* (**D**), and *hsa-miR-27b-3p* (**E**) between LUAD and normal lung tissues. (**F**–**J**) Survival analysis of *hsa-miR-1-3p* (**F**), *hsa-miR-26a-5p* (**G**), *hsa-miR-664b-3p* (**H**), *hsa-miR-135b-5p* (**I**), and *hsa-miR-27b-3p* (**J**) in TCGA-LUAD cohort.

**Figure 6 biomedicines-12-01477-f006:**
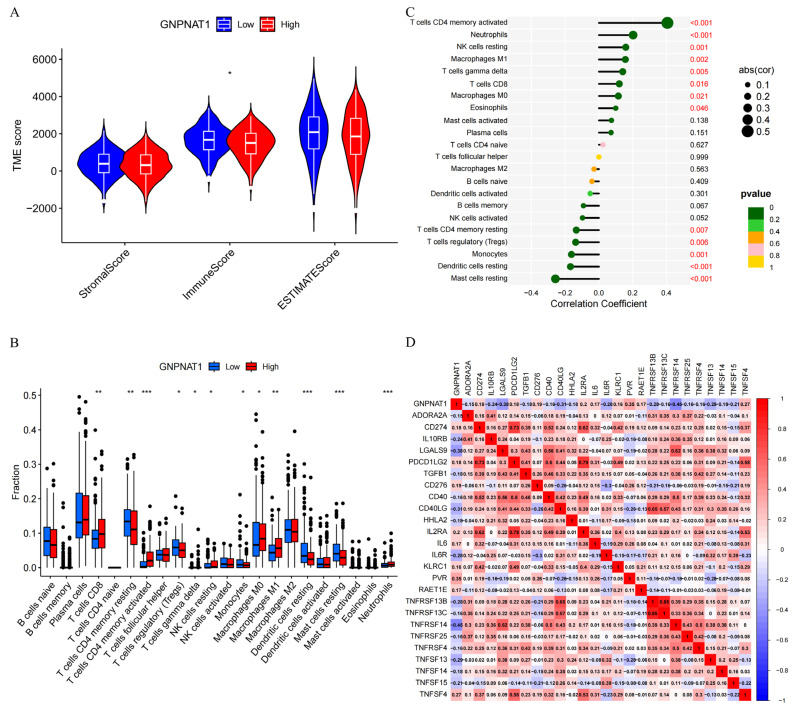
Correlation analysis of *GNPNAT1* and immune infiltration, immunomodulator genes in LUAD. (**A**) Stromal scores, immune scores, ESTIMATE scores between high-*GNPNAT1*-expression group and low-*GNPNAT1*-expression group. (**B**,**C**) CIBERSORT analysis between the high-*GNPNAT1* and low-*GNPNAT1* group. (**C**) Correlation analysis of *GNPNAT1* expression and immune infiltration. (**D**) Correlation analysis of *GNPNAT1* and immunomodulator genes. * *p* < 0.05, ** *p* < 0.01, *** *p* < 0.001.

**Figure 7 biomedicines-12-01477-f007:**
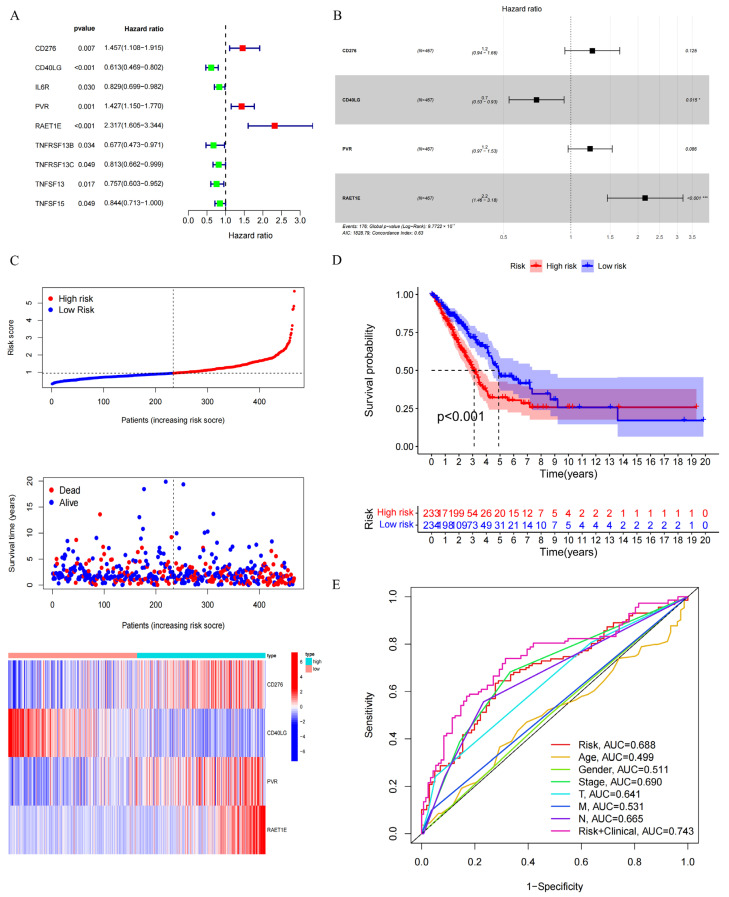
Construction of the 4-immunomodulator genes’ signature in TCGA-LUAD cohort. (**A**) Univariate analysis of *GNPNAT1*-associated immunomodulator genes. (**B**) Development of the 4-mmunomodulator gene signature based on *GNPNAT1*-associated immunomodulator genes in TCGA-LUAD cohort. (**C**) Risk score distribution of LUAD patient, survival status analysis of LUAD patients, and heatmap of the 4-mmunomodulator gene expression. (**D**) Survival analysis for the 4-mmunomodulator gene in TCGA-LUAD cohort. (**E**) ROC analysis for the 4-mmunomodulator gene in TCGA-LUAD cohort. * *p* < 0.05, *** *p* < 0.001.

**Figure 8 biomedicines-12-01477-f008:**
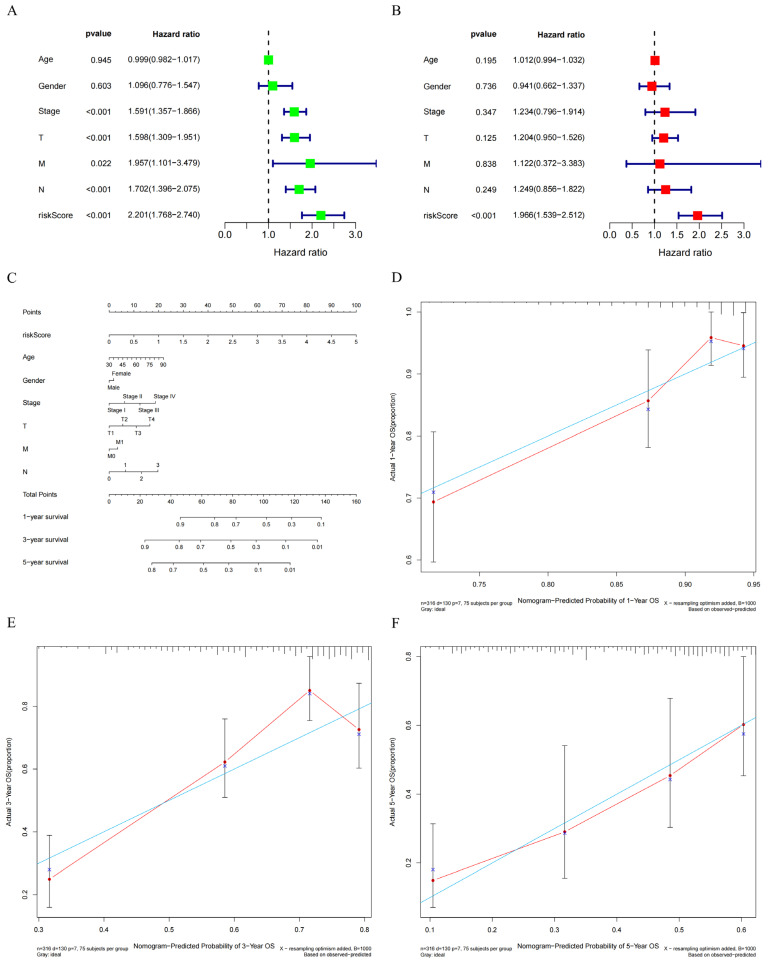
The independent prognosis analysis for the 4-immunomodulator genes’ signature in TCGA-LUAD cohort. (**A**,**B**) Univariate and multivariate Cox regression analyses for the 4-mmunomodulator gene signature. (**C**) Construction of a nomogram based on the 4-mmunomodulator gene signature in the TCGA cohort. (**D**–**F**) Calibration curves of nomogram for the signature at 1, 3, and 5 years.

**Figure 9 biomedicines-12-01477-f009:**
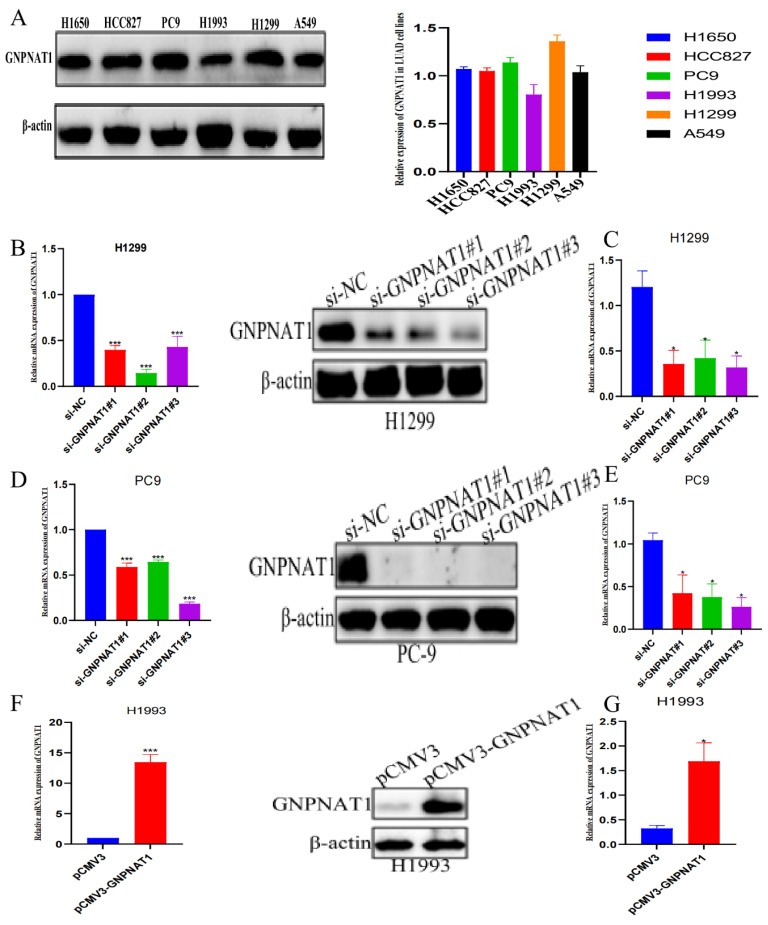
Expression level of GNPNAT1 in LUAD cell lines. (**A**) Western blot analysis of *GNPNAT1* expression in six LUAD cell lines. (**B**,**C**) qRT-PCR and Western blotting detected the efficiency of pCMV3-*GNPNAT1* transfection in H1993. (**D**,**E**) Expression level of *GNPNAT1* from H1299 transfected with si-*GNPNAT1* were assessed using qRT-PCR and Western blotting. (**F**,**G**) Expression level of *GNPNAT1* from PC9 transfected with si-GNPNAT1 were assessed using qRT-PCR and Western blotting. Expression level of *GNPNAT1* were normalized to *GAPDH* in qRT-PCR, while expression level of *GNPNAT1* were normalized to β-actin in Western blotting. Data were presented as mean ± SEM, *n* = 3. * *p* < 0.05, *** *p* < 0.001.

**Figure 10 biomedicines-12-01477-f010:**
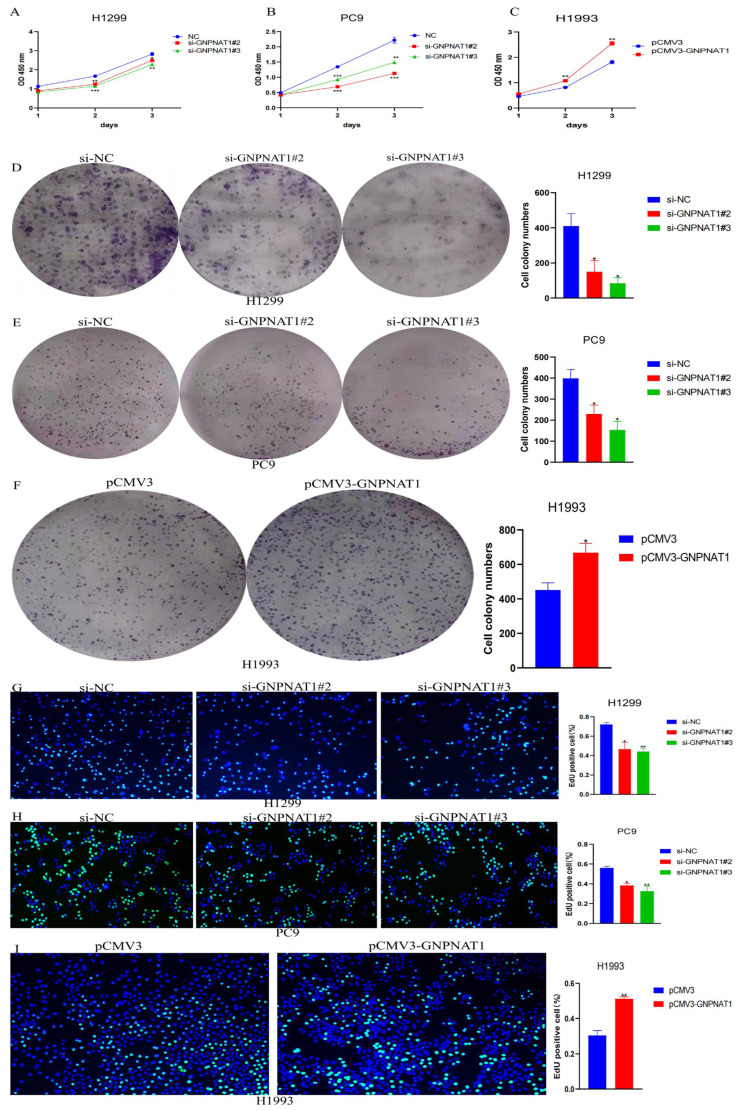
The effect of *GNPNAT1* on LUAD cells proliferation. (**A**–**C**) CCK8 assay was executed to detect the proliferative ability of H1299 (**A**) or PC9 (**B**) transfected with si-*GNPNAT1* and H1993 (**C**) transfected with pCMV3-*GNPNAT1*. (**D**–**F**) Colony formation assay was performed to test the proliferative ability of H1299 (**D**) or PC9 (**E**) transfected with si-*GNPNAT1* and H1993 (F) transfected with pCMV3-*GNPNAT1*. (**G**–**I**) EdU assay was utilized to validate the proliferative ability of H1299 (**G**) or PC9 (**H**) transfected with si-*GNPNAT1* and H1993 (**I**) transfected with pCMV3-*GNPNAT1*. Data derived from three independent experiments were presented as mean ± SEM, *n* = 3. * *p <* 0.05, ** *p <* 0.01, *** *p* < 0.001.

**Figure 11 biomedicines-12-01477-f011:**
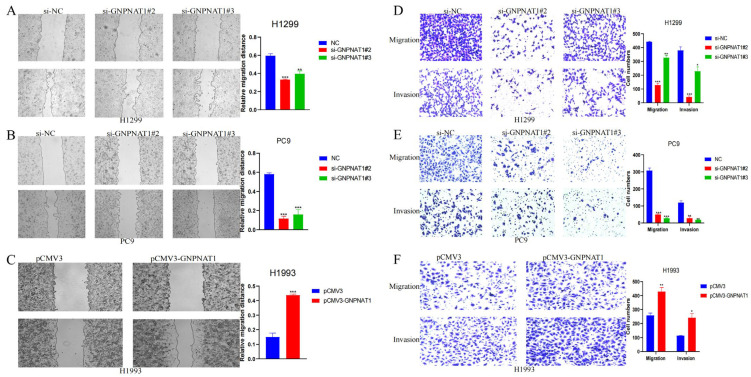
The effect of *GNPNAT1* knockdown or overexpression on migration, invasion of LUAD cells. *GNPNAT1* knockdown inhibited H1299 (**A**,**D**) or PC9 (**B**,**E**) cell migration and invasion ability. H1299 or PC9 cell transfected with si-NC or si-*GNPNAT1* were seeded in a transwell chamber with or without Matrigel coating at a ratio of 3 × 10^4^ cells/well, followed by incubation for 24 h. The cell numbers for 0 h or 24 h were quantified by Fiji. *GNPNAT1* overexpression promoted H1993 (**C**,**F**) cell migration and invasion ability. H993 cell transfected with pCMV3 or pCMV3-*GNPNAT1* were seeded in a transwell chamber with or without Matrigel coating at a ratio of 3 × 10^4^ cells/well, followed by incubation for 24 h. The cell numbers for 0 h or 24 h were quantified by Fiji. Data were presented as mean ± SEM. * *p <* 0.05, ** *p <* 0.01, *** *p <* 0.001.

**Figure 12 biomedicines-12-01477-f012:**
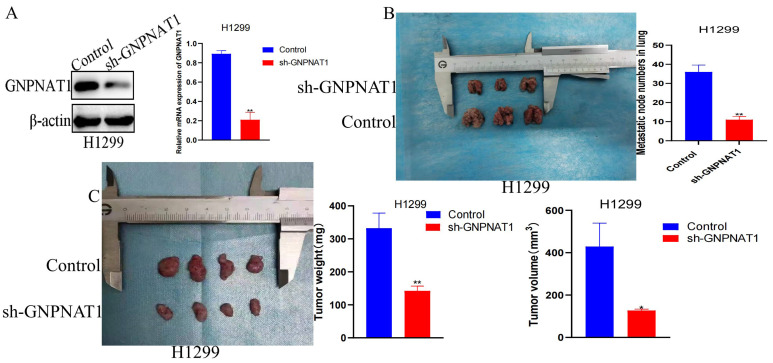
The effect of *GNPNAT1* knockdown on LUAD cell metastasis. (**A**) Expression levels of *GNPNAT1* were detected via Western blotting in H1299 cell stably transfected with sh-*GNPNAT1*. (**C**) H1299 cell stably transfected with sh-*GNPNAT1* were subcutaneously injected in the right subaxillary of BALB/c nude mice. The volume and weight of tumor xenograft from sh-*GNPNAT1* were lower than that from control. (**B**) *GNPNAT1* knockdown inhibited H1299 cell metastasis in vivo. Data were presented as mean ± SEM. * *p* < 0.05, ** *p* < 0.01.

**Figure 13 biomedicines-12-01477-f013:**
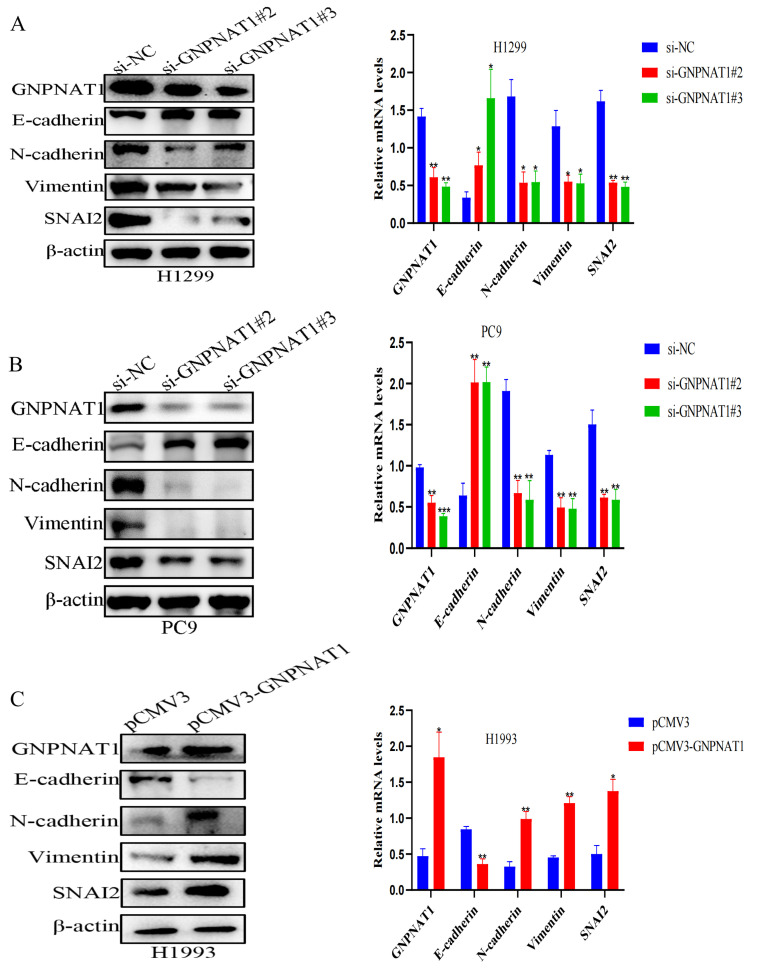
*GNPNAT1* knockdown suppressed EMT in LUAD cells. (**A**,**B**) Western blotting confirmed that *GNPNAT1* knockdown significantly resulted in loss of *GNPNAT1*, N-cadherin, Vimentin, and Snai2 expression levels, while markedly enhanced expression of E-cadherin in H1299 and PC9, respectively. (**C**) *GNPNAT1* overexpression significantly promoted *GNPNAT1*, N-cadherin, Vimentin, and Snai2 expression, while markedly inhibited expression of E-cadherin in H993. Relative expression levels of *GNPNAT1*, E-cadherin, N-cadherin, Vimentin, and Snai2 were normalized against β-actin. Data were presented as mean ± SEM, *n* = 3. * *p* < 0.05, ** *p* < 0.01, *** *p* < 0.001.

**Figure 14 biomedicines-12-01477-f014:**
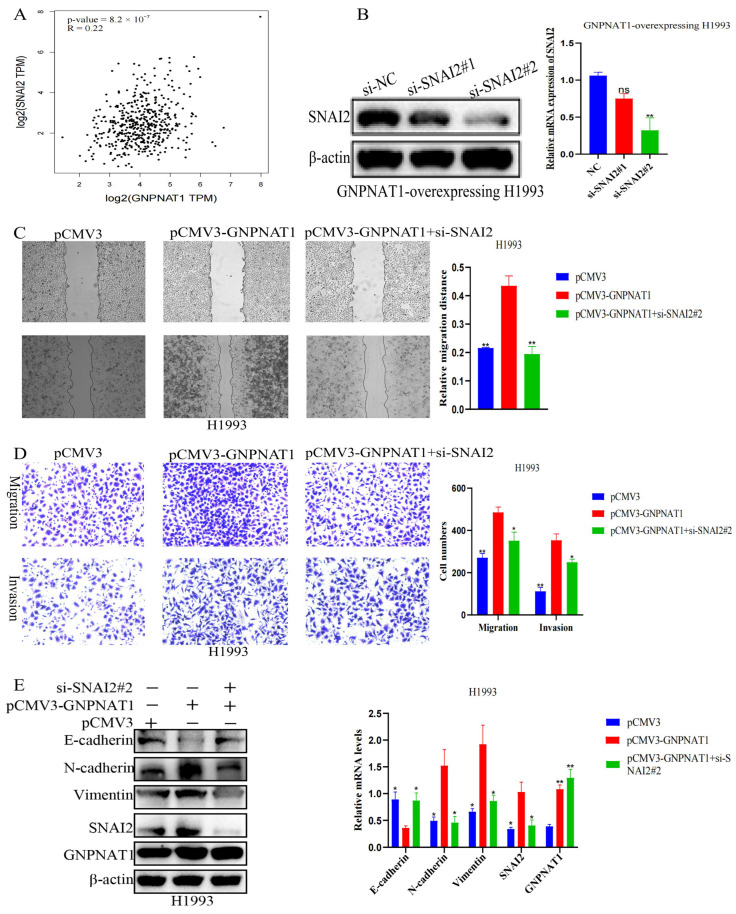
Snai2 was required in *GNPNAT1*-mediated EMT. (**A**) The correlation between Snai2 and *GNPNAT1* was identified in GEPIA2 (http://gepia2.cancer-pku.cn/, accessed on 6 January 2023). (**B**) Western blotting detected the expression level of *GNPNAT1* in *GNPNAT1*-overexpressing H1993 transfected with si-*Snai2*. Relative expression levels of Snai2 were normalized against β-actin in *GNPNAT1*-overexpressing H1993 cells. (**C**,**D**) Snai2 knockdown repressed *GNPNAT1*-mediated migration and invasion in H1993 cells. (**E**) Snai2 knockdown inhibited *GNPNAT1*-mediated EMT in H1993 cells. Relative expression levels of *GNPNAT1*, E-cadherin, N-cadherin, Vimentin, and Snai2 were normalized against beta actin. Data were presented as mean ± SEM, *n* = 3. ns: no significance, * *p* < 0.05, ** *p* < 0.01.

**Figure 15 biomedicines-12-01477-f015:**
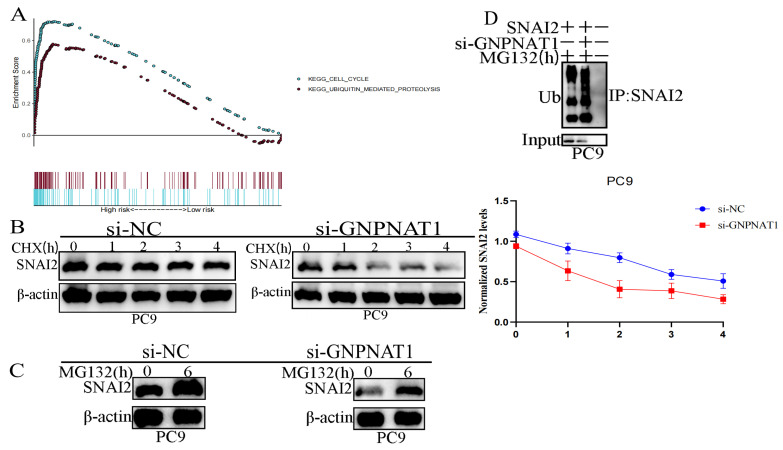
*GNPNAT1* maintained stability of Snai2 protein via inhibiting ubiquitination and degradation of Snai2 protein. (**A**) Functional analysis of *GNPNAT1* via GSEA. (**B**) PC9 cell transfected with si-*GNPNAT1* or si-NC were treated with CHX at 0, 1, 2, 3, 4 h, respectively. Western blot was performed to detect *Snai2* expression levels in the treated PC9 cells, and the half-life of Snai2 was analyzed in PC9 cell transfected with si-*GNPNAT1* or si-NC. (**C**) PC9 cell transfected with si-*GNPNAT1* or si-NC were treated with MG132 at 0 and 6 h, respectively. *Snai2* expression levels were detected using Western blot. (**D**) Snai2 ubiquitination levels were detected through immunoprecipitation using the anti-*Snai2* antibody in PC9 cell transfected with si-*GNPNAT1* or si-NC.

## Data Availability

RNA data are available at the TCGA database and GEO datasets (accession numbers: GSE32863, GSE40791, GSE75037, GSE115002, GSE13213, and GSE72094). The data that support the findings of this study are available from the corresponding author upon reasonable request.

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
