# Peer review of "GNPNAT1 Serves as a Prognostic Biomarker Correlated with Immune Infiltration and Promotes Cancer Cell Metastasis through Stabilization of Snai2 in Lung Adenocarcinoma"

_biomedicines, 2024, doi:10.3390/biomedicines12071477_

Round 1

Reviewer 1 Report

Comments and Suggestions for Authors

In this study, the authors attempt to investigate the role of GNPNAT1 in immune infiltration and cancer metastasis of lung adenocarcinoma. The authors successfully demonstrate the role of GNPNAT1 in cell proliferation, migration, and metastasis of lung adenocarcinoma. The insight of this study is novel and attractive for the translational study field of the lung adenocarcinoma. The following are my comments.

1)     Page 5 Line 194, 198. Please keep the symbol multiply consistent (X or *)

2)     Page 5 Line 198. The standard calculation formulae in measuring xenograft tumor volume is “V= 0.5x Length x Width 2” Please correct the result accordingly.

3)     Page 8 Line 265 - 268. Please rewrite the statement.

4)     Page 10 Line 331-332. Please perform the statistic analysis for the comparison of GNPNATI expression and report the P-value.  

Author Response

Dear reviewer 1:

Thank you for your decision and constructive comments on my manuscript. We have carefully considered the suggestion of reviewer and we have tried our best to improve and made some changes in the manuscript. The red part in the manuscript is the revised parts. Revision notes, point-to-point, are given as follows:

Reviewer 1:

Comment 1: Page 5Line 194,198. Please keep the symbol multiply consistent (X or*)

Response: The symbol multiple has been adjusted to be consistent in the manuscript.

Comment 2: Page 5 Line 198. The standard calculation formulae in measuring xenograft tumor volume is"V=0.5x Length x Width 2" Please correct the result accordingly.

Response: We feel sorry for our carelessness. As it is suggested by the Reviewer,we have corrected "V= Length x Width 2" in line 198 on page 5 into "V=0.5x Length x Width 2".

Comment 3: Page 8 Line 265-268.Please rewrite the statement.

Response: Thank you for this valuable feedback. The statement in line 265-268 on page 8 has been rewritten and re-revised.

Comment 4: Page 10 Line 331-332.Please perform the statistic analysis for the comparison of GNPNATl expression and report the P-value.

Response: We respectfully disagree. In this study, the cell lines we used, including H1650, HCC827, PC9, H1993, H1299, and A549, are all lung adenocarcinoma cell lines. We only need to detect the expression of GNPNAT1 in these cell lines. The results indicate that GNPNAT1 has the highest expression level in PC9 and H1299 cells, and the lowest expression level in H1993 cells. H1299 and PC9 were selected to perform knockdown, and H1993 was selected to perform over-expression. Therefore, it is not necessary for performing statistical analysis.

We sincerely appreciate your thorough and thoughtful review which has undoubtedly enhanced the quality of this manuscript. We hope that the revision we made adequately addressed your concerns. If you have any further suggestions, please do not hesitate to let us know.

Once again, thank you for your time and expertise in reviewing our manuscript.

Best regard,

YanYu

Harbin Medical University Cancer Hospital

Reviewer 2 Report

Comments and Suggestions for Authors

This is an interesting research study with adequate novelty and quality. However. several points should be addressed.

Abstract

- Subheadings (Background, Methis, etc) should be included into the abstract.

- The authors used several abbreviations which should be explained, e.g  LUAD is firstly reported in line 11 but this abbreviation is explianed in line 19. What the other abbreviations mean?

- In line 12, the authors could simply report that ".....LUAD metastasis remain still unknown.".

- The sentence in lines 21-25 is quite long and should be split into two sentences.

Introduction

- The last paragraph in lines 57-66 reported the results of the study. These should be reported in the Results and Discussion sections but not in the Introduction section. Instead of the above, the authors should describe at the end of the Introduction section the literature gap that it exists concerning the topic of their study. Moreove, in this point, the authors should report the aim of their study which will cover the above literature gap.

Materials and Methods

- The authors should report the relevant information concerning ethical concerns and their approval in this point.

- The authors should try to describe more analytically the statistical analysis methods used.

Results

- Figure 2 is very complex and should be split at least into two separate figures by improving their resolution as well.

- Again, Figure 3 is very complex and should be split at least into two separate figures by improving their resolution as well.

- Again, Figure 4 is very complex and should be split at least into two separate figures by improving their resolution as well.

- Again, Figure 7 is very complex and should be split at least into two separate figures by improving their resolution as well.

Discussion

- This section is well-written and well-organized. However, English editing language is highly recommended.

Conclusion

- In this section the authors should report what future studies could be performed based on their results and their expertise.

References

- Some references published in the last 2-3 years should be included.

Comments on the Quality of English Language

Extensive editing of English language required

Author Response

Dear reviewer 2:

Thank you for your decision and constructive comments on my manuscript. We have carefully considered the suggestion of reviewer and we have tried our best to improve and made some changes in the manuscript. The red part in the manuscript is the revised parts. Revision notes, point-to-point, are given as follows:

Reviewer 2

Abstract

Comment 1: Subheadings (Background,Methis,etc) should be included into the abstract.

Response: Subheadings, including Background, Methods, Results, Conclusion, are added to the abstract of the manuscript.

Comment 2: The authors used several abbreviations which should be explained,e.g LUAD is firstly reported in line 11 but this abbreviation is explained in line 19. What do the other abbreviations mean?

Response: The abbreviations in the abstract of the manuscript are explained, the problem"LUAD is firstly reported in line 11 but this abbreviation is explained in line 19" are corrected.

Comment 3: In line 12, the authors could simply report that"LUAD metastasis remain still unknown."

Response: Thank you for this valuable feedback. “the exact role and related mechanism of GNPNAT1 in regulating LUAD metastasis remain largely unknown” is corrected into “the exact role and related mechanism of GNPNAT1 in LUAD metastasis remain unknown”.

Comment 4: The sentence in lines 21-25 is quite long and should be split into two sentences.

Response: Thank you for this valuable feedback. The sentence in lines 21-25 is divided into two sentences.

Introduction

Comment 5: The last paragraph in lines 57-66 reported the results of the study. These should be reported in the Results and Discussion sections but not in the Introduction section. Instead of the above, the authors should describe at the end of the Introduction section the literature gap that it exists concerning the topic of their study. Moreover, in this point, the authors should report the aim of their study which will cover the above literature gap.

Response: Thank you for your question. The last paragraph in the introduction of the manuscript is rewritten.

Materials and Methods

Comment 6: The authors should report the relevant information concerning ethical concerns and their approval in this point.

Response: The ethical concerns and relevant approval for this study is added to in vivo assay part of Materials and Methods in the manuscript.

Comment 7: The authors should try to describe more analytically the statistical analysis methods used.

Response: The more statistical analysis methods are added to statistical analysis part of Materials and Methods in the manuscript.

Results

Comment 8: Figure 2 is very complex and should be split at least into two separate figures by improving their resolution as well.

Response: Thank you for this valuable feedback. The old Figure 2 has been split into Figure 2 and Figure 3.

Fig. 2 GNPNAT1 was associated with poor prognosis in LUAD. (A-C) Data from GEO and TCGA database demonstrated that GNPNAT1 high expression had a worse OS than GNPNAT1 low expression using survival analysis. (D) Data from TCGA indicated that GNPNAT1 high expression were positively correlated with T-stage, TNM stage, lymph node metastasis.

Fig. 3 GNPNAT1 was an independent prognostic factor for LUAD patients. (A) The univariate Cox proportional hazard model analysis. (B) The multivariate Cox proportional hazard model analysis. (C, D) Development of a nomogram model on the basis of GNPNAT1 and calibration of nomogram for GNPNAT1 at 1, 3, 5 years in TCGA-LUAD cohort.

Comment 9: Again,Figure 3 is very complex and should be split at least into two separate figures by improving their resolution as well.

Response: Thank you for this valuable feedback. The old Figure 3 has been split into Figure 4 and Figure 5.

Fig. 4 The potential biological functions and miRNA targets of GNPNAT1 in LUAD. (A) The GNPNAT1-associated genes in TCGA-LUAD cohort. (B, C) The Go and KEGG analysis of the GNPNAT1-associated genes in TCGA-LUAD cohort. (D) The miRNAs interacted with GNPNAT1 (https://rnasysu.com/encori/index.php). (E-I) Correlation analysis between GNPNAT1 and hsa−miR−1−3p (E), hsa−miR−26a−5p (F), hsa−miR−664b−3p (G), hsa−miR−135b−5p (H), and hsa−miR−27b−3p (I) in LUAD.

Fig. 5 The differential expression and survival analysis for GNPNAT1-associated miRNA targets. (A-E) Differential expression analysis of hsa−miR−1−3p (A), hsa−miR−26a−5p (B), hsa−miR−664b−3p (C), hsa−miR−135b−5p (D), and hsa−miR−27b−3p (E) between LUAD and normal lung tissues. (F-J) Survival analysis of hsa−miR−1−3p (F), hsa−miR−26a−5p (G), hsa−miR−664b−3p (H), hsa−miR−135b−5p (I), and hsa−miR−27b−3p (J) in TCGA-LUAD cohort.

Comment 10: Again, Figure 4 is very complex and should be split at least into two separate figures by improving their resolution as well.

Response: Thank you for this valuable feedback. The old Figure 4 has been split into Figure 6, Figure 7 and Figure 8.

Fig. 6 Correlation analysis of GNPNAT1 and immune infiltration, immunomodulator genes in LUAD. (A) Stromal scores, Immune scores, ESTIMATE scores between high GNPNAT1 expression group and low GNPNAT1 expression group. (B) (E) CIBERSORT analysis between the high GNPNAT1 and low GNPNAT1 group. (C) Correlation analysis of GNPNAT1 expression and immune infiltration. (D) Correlation analysis of GNPNAT1 and immunomodulator genes.

Fig. 7 Construction of the 4-immunomodulator genes signature in TCGA-LUAD cohort. (A) Univariate analysis of GNPNAT1 associated immunomodulator genes. (B) Development of the 4-mmunomodulator gene signature based on GNPNAT1 associated immunomodulator genes in TCGA-LUAD cohort. (C) Risk score distribution of LUAD patient, survival status analysis of LUAD patients, and heatmap of the 4-mmunomodulator gene expression. (D) Survival analysis for the 4-mmunomodulator gene in TCGA-LUAD cohort. (E) ROC analysis for the 4-mmunomodulator gene in TCGA-LUAD cohort.

Fig. 8 The independent prognosis analysis for the 4-immunomodulator genes signature in TCGA-LUAD cohort. (A, B) Univariate and multivariate Cox regression analyses for the 4-mmunomodulator gene signature. (C) Construction of a nomogram based on the 4-mmunomodulator gene signature in the TCGA cohort. (D-F) Calibration curves of nomogram for the signature at 1, 3, and 5-years.

Comment 11: Again,Figure 7 is very complex and should be split at least into two separate figures by improving their resolution as well.

Response: Thank you for this valuable feedback. The old Figure 10 has been split into Figure 11 and Figure 12.

 Fig. 11 The effect of GNPNAT1 knockdown or overexpression on migration, invasion of LUAD cells. GNPNAT1 knockdown inhibited H1299 (A, D) or PC9 (B, E) cell migration and invasion ability. H1299 or PC9 cell transfected with si-NC or si-GNPNAT1 were seeded in transwell chamber with or without matrigel-coated at a ratio of 3 x 104 cells/well, followed by incubation for 24h. The cell numbers for 0h or 24h were quantified by Fiji. GNPNAT1 overexpression promoted H1993 (C, F) cell migration and invasion ability. H993 cell transfected with pCMV3 or pCMV3-GNPNAT1 were seeded in transwell chamber with or without matrigel-coated at a ratio of 3 x 104 cells/well, followed by incubation for 24h. The cell numbers for 0h or 24h were quantified by Fiji. Data were presented as mean±SEM. *P<0.05, **P<0.01, ***P<0.001.

Fig. 12 The effect of GNPNAT1 knockdown on LUAD cells metastasis. (A) Expression levels of GNPNAT1 were detected via western blotting in H1299 cell stably transfected with sh-GNPNAT1. (C) H1299 cell stably transfected with sh-GNPNAT1 were subcutaneously injected in the right subaxillary of BALB/c nude mice. The volume and weight of tumor xenograft from sh-GNPNAT1 were lower than that from control. (B). GNPNAT1 knockdown inhibited H1299 cell metastasis in vivo. Data were presented as mean±SEM. *P<0.05, **P<0.01.

Discussion

Comment 12: This section is well-written and well-organized. However, English editing language is highly recommended.

Response: We apologized for the poor language of our manuscript. We worked on the manuscript for a long time and the repeated addition and removal of sentences and sections obviously led to poor readability and have also involved native English speaker for language corrections. We really hope that the flow and language level have been substantially improved.

Conclusion

Comment 12: In this section the authors should report what future studies could be performed based on their results and their expertise.

Response: Thank you for your comment. The section is rewritten.

References

Comment 13: Some references published in the last 2-3 years should be included.

Response: We sincerely thank the reviewer fore careful reading. At present, there is still limited research on GNPNAT1 in malignant tumors. Some references published in the last 2-3 years is cited in this study.

We sincerely appreciate your thorough and thoughtful review which has undoubtedly enhanced the quality of this manuscript. We hope that the revision we made adequately addressed your concerns. If you have any further suggestions, please do not hesitate to let us know.

Once again, thank you for your time and expertise in reviewing our manuscript.

Best regard,

YanYu

Harbin Medical University Cancer Hospital

Reviewer 3 Report

Comments and Suggestions for Authors

This study analyzed the role of GNPNAT1 in lung adenocarciona (LUAD). The study is mixed bioinformatics and also invitro/in vivo modeling. GNPNAT1 appears to be a poor prognostic marker in LUAD.

 Comments:

 (1) Line 43. Regarding LUAD dataset.

Could you please add the website link “https://portal.gdc.cancer.gov/projects/TCGA-LUAD”?

 (2) Lines 47-50. Could you please show the function of GNPNAT1 in a figure?

 (3) Is GNPNAT1 a metabolism key player?

 (4) Lines 69-75. Several datasets are shown. Please add the corresponding references for each dataset as shown in the GEO website to acknowledge the authors of the datasets.

 (5) Lines 69-75. What are the clinicopathological characteristics of the different datasets that were used? Is it possible to make a table that could be shown in the appendix?

 (6) Lines 77-83. Please add the references and website link for the R packages (estimate, cibersort, etc.)

 (7) Line 86. Could you please add the address and website of the “Heilongjiang Cancer Institute”?

 (8) Line 85-86. Could you please add a table showing the properties of each cell line that is mentioned?

 (9) In Material and Methods, please add the catalog number and company name and address of each reagent that is mentioned.

 (10) Line 92. Do you know the brand and model of the humidified incubator?

 (11) Please write gene names in italics.

 (12) Line 117, regarding “cultured LUAD cell lines”. I understand that you mean lung adenocarcinoma cell lines. However, they have nothing to do with the LUAD dataset. Should you better use a different name?

 (13) Line 126. What was the secondary antibody?

 (14) What was the chemiluminescence reagent?

 (15) Line 166. Citrate buffer was used as antigen retrieval. Could you please explain in more details the temperature, time, etc.?

 (16) Line 174. Is this type of quantification a “H-score” method?

 (17) Line 208. Why using the standard error of the mean instead of standard deviation?

 (18) Line 193. Why BALB/c mice were used instead of NOD/SCID?

 (19) In control tissue, what is the internal control of IHC in Figure 1G?

 (20) In Figure 2A-C, the OS is different and statistically significant. However, the lines are quite close. From a clinical point of view, is this difference relevant?

 (21) Line 248. The correlation r value is 0.55. Is this a “low” correlation value?

 (22) In Figure 4. What was the correlation between GNPNAT1 and PD-L1?

 (23) Since WB are cropped. Please confirm that original images are available and provide same result.

 (24) In Figure 8. What is the difference between e-cadherin and n-cadherin?

 (25) Figure 9B and E is very small and difficult to read in pdf file.

 (26) In 9B, are 3 signals of vimentin equal?

 (27) In Figure 10A. What are the genes of the leading edge?

 (28) In Figure 10, the size of letters is too variable, some very big, others very small. Please improve the quality of the figures of the manuscript.

 (29) By gene expression, what is the bivariate correlation between GNPNAT1 and Slug?

 (30) Should you use SNAI2 instead of SLUG?

Author Response

Dear reviewer 3:

Thank you for your decision and constructive comments on my manuscript. We have carefully considered the suggestion of reviewer and we have tried our best to improve and made some changes in the manuscript. The red part in the manuscript is the revised parts. Revision notes, point-to-point, are given as follows: 

Comment 1: Line 43. Regarding LUAD data set. Could you please add the website link“htps://portal.gdc.cancer.gov/projects/TCGA-LUAD"?

Response: Thanks for your suggestion. The website link“htps://portal.gdc.cancer.gov/projects/TCGA-LUAD" in Line is added.

Comment 2: Lines 47-50. Could you please show the function of GNPNAT1 in a figure?

Response: Thank you for this valuable feedback. The function of GNPNAT1 is showed in supplementary Figure 1.

Supplementary Fig. 1 The detail function of GNPNAT1.

Comment 3: Is GNPNAT1 a metabolism key player?

Response: Thanks. GNPNAT1 is the rate-limiting enzyme in the second step of the HBP (hexosamine biosynthetic pathway) process and convert glucosamine-6-phosphate to N-acetylglucosamine-6-phosphate. Therefore, GNPNAT1 is considered as a metabolism key player.

Comment 4: Lines 69-75. Several datasets are shown. Please add the corresponding references for each data set as shown in the GEO website to acknowledge the authors of the datasets.

Response: Thanks for your comment. The corresponding references for each data set are added.

Comment 5: Lines 69-75. What are the clinicopathological characteristics of the different datasets that were used? Is it possible to make a table that could be shown in the appendix?

Response: Thank you for this valuable feedback. The clinicopathological characteristics of the different datasets are presented in supplementary table 4 and table 5.

Supplementary Table 1. Clinicopathological characteristics of LUAD patients 

Characteristics

TCGA-LUAD cohort (n=468)

GSE72094 (n=442)

GSE13213 (n=117)

Age (y)

≤65

227(46.71%)

127(28.73%)

78(66.67%)

>65

240(49.38%)

294(66.52%)

39(33.33%)

unknown

19(3.91%)

21(4.75%)

-

Gender

Male

264(54.32%)

240(54.30%)

57(48.72%)

Female

222(45.68%)

202(45.70%)

60(51.28%)

Tumor stage

T1+T2

423(87.04%)

-

-

T3+T4

60(12.34%)

-

-

unknown

3(0.62%)

-

-

Lymph node stage

N0+N1

402(82.72%)

-

-

N2+N3

72(14.81%)

-

-

unknown

12(2.47%)

-

-

Distant mentastasis stage

M0

333(68.52%)

-

-

M1

24(4.94%)

-

-

unknown

129(26.54%)

-

-

Pathological stage

I+II

374(76.95%)

334(75.57%)

92(78.63%)

III+IV

104(21.40%)

80(18.10%)

25(21.37%)

unknown

8(1.65%)

28(6.33%)

-

Survival status

Alive

304(62.55%)

298(67.42%)

68(58.12%)

Deceased

182(37.45%)

122(27.60%)

49(41.88%)

Unknown

-

22(4.98%)

-

Follow-up(y)

< 5

426(87.65%)

393(88.91%)

42(35.90%)

≥5

51(10.49%)

5(1.13%)

75(64.10%)

Unknown

9(1.85%)

44(9.95%)

-

Supplementary Table 2. Clinicopathological characteristics of LUAD patients

Characteristics

GSE32863

GSE40791

GSE75037

GSE115002

Histology

Non-malignant lung

58

100

83

52

Lung adenocarcinoma

58

94

83

52

Age(y)

≤65

19

-

29

39

>65

39

-

54

13

Gender

Male

13

41

24

26

Female

45

53

59

26

Smoking

Yes

29

80

53

9

No

29

4

30

43

unknown

-

10

-

-

Clinical stage

I+II

45

92

70

26

III+IV

13

2

13

26

Comment 6: Lines 77-83. Please add the references and website link for the R packages(estimate,cibersort,etc.)

Response: Thank you for this valuable feedback. The references and website link for the R packages (estimate,cibersort,etc.) in Line 77-83 are added.

Comment 7: Line 86.Could you please add the address and website of the “Heilongjiang Cancer Institute"?

Response: Thank you for this valuable feedback. The address and website of the “Heilongjiang Cancer Institute" in Line 86 is added.

Comment 8: Line 85-86. Could you please add a table showing the properties of each cell line that is mentioned?

Response: We respectfully disagree. “The human lung adenocarcinoma cell lines PC9, H1299, A549, H1993, HCC827, and H1650 were preserved in the Heilongjiang Cancer Institute” indicates that the properties of these cell lines belong to LUAD cell lines. Therefore, we believe it is not necessary to add a table showing the properties of each cell line.

Comment 9: In Materia l and Methods, please add the catalog number and company name and address of each reagent that is mentioned.

Response: Thanks for your question. The catalog number and company name and address of each reagent is added in the manuscript.

Comment 10: Line 92. Do you know the brand and model of the humidified incubator?

Response: We sincerely thank the reviewer for careful reading. In this study, the humidified incubator used in cell culture is ThermoFisher Scientific CO2 incubator, which can maintain constant humidity, humidity and control CO2 concentration.

Comment 11: Please write gene names in italics.

Response: Thanks. The question is addressed.

Comment 12: Line 117, regarding "cultured LUAD cell lines".I understand that you mean lung adenocarcinoma cell lines. However,they have nothing to do with the LUAD dataset. Should you better use a different name?

Response: Thanks for your careful check. "cultured LUAD cell lines" in Line 117 is deleted. 

Comment 13: Line 126.What was the secondary antibody?

Responce: Thanks. The secondary antibody is HRP Goat anti-mouse IgG or HRP Goat anti-rabbit IgG

Comment 14: What was the chemiluminescence reagent?

Response: Thanks for your suggestion. The chemiluminescence reagent is  Chemistar TM High-sig ECL Western Blotting Substrate.

Comment 15: Line 166.Citrate buffer was used as antigen retrieval.Could you please explain in more details the temperature,time,etc.?

Response: Thank you for this valuable feedback. The section soaked in 0.01mmol lemon salt solutio is heated for 2’30’’ in the microwave oven. Finally, the section is further heated for 5’ in a pressure cooker with keeping the pressure between 0.8 and 1.0Mpa.

Comment 16: Line 174.Is this type of quantification a“H-score”method?

Response: Thank you for this valuable feedback. Based on research by Song et al, we believe that this type of quantification is a“H-score”method (Journal of Experimental & Clinical Cancer Research, DOI 10.1186/s13046-016-0427-7.).

Comment 17: Line 208.Why using the standard error of the mean instead of standard deviation?

Response: We sincerely thank the reviewer for this valuable feedback. The standard error of the mean is used to measure the estimation accuracy of sample statistics and is suitable for describing the reliability of samples. Therefore, we use standard error of the mean instead of standard deviation.

Comment 18: Line 193.Why BALB/c mice were used instead of NOD/SClD?

Response: Thank you for this valuable feedback. BALB/c nude mice is utilized to conduct in vivo assays to be contributed to convenient operation and observation.

Comment 19: In control tissue,what is the internal control of IHC in Figure 1G?

Response: Thanks for your question. The section without GNPNAT1 expression is identified as internal controls in control tissue.

Comment 20: In Figure 2A-C, the OS is different and statistically significant.However,the lines are quite close. From a clinical point of view,is this difference relevant?

Response: Thanks for your suggestion. Based on our research finding, we believe that this difference is relevant from a clinical point view.

Comment 21: Line 248.The correlation r value is 0.55. Is this a"low" correlation value?

Response: Thanks for your question. The correlation relationship can be divided into no correlation, weak correlation, moderate correlation, strong correlation, and extremely strong correlation. “0.4< r <0.6” is defined as moderate correlation. Therefore, “r = 0.55” is not a low correlation but a moderate correlation.

Comment 22: In Figure 4. What was the correlation between GNPNAT1 and PD-L1?

Response: we observe a positive correlation between GNPNAT1 and PD-L1 with Spearman's r = 0.18.

Comment 23: Since WB are cropped. Please confirm that original images are available and provide same result.

Response: Thank you for your reminding. The original images for WB are presented as followed:

Comment 24:  In Figure 8.What is the difference between e-cadherin and n-cadherin?

Response: Thanks for your question. EMT plays a critical role in tumor metastasis. EMT is featured by the loss of epithelial marker (E-cadherin) and up-regulation of mesenchymal markers (Vimentin and N-cadherin).

Comment 25: Figure 9B and E is very small and difficult to read in pdf file.

Response: Thanks for your comment. Figure 9B and E has been enlarged so that they are easy to read in PDF file.

Comment 26: In 9B,are 3 signals of vimentin equal?

Response: Thanks. Figure 9B shows the expression levels of si-SNAI2-1 and si-SNAI2-2 in GNPNAT1-overexreing H1993 cells. si-SNAI2-2 was successfully knocked down in GNPNAT1-overexreing H1993 cells., while si-SNAI-1 was not knocked down in GNPNAT1-overexreing H1993 cells.

Comment 27: In Figure 10A.What are the genes of the leading edge?

Response: Thank you for this valuable feedback. In this study, we mainly focus on the impact of GNPNAT1 on lung adenocarcinoma metastasis, and the leading edge in Figure 10A provides new clues for our future research on the mechanism of lung adenocarcinoma metastasis.

Comment 28: In Figure 10,the size of letters is too variable, some very big,others very small.Please improve the quality of the figures of the manuscript.

Response: Thank you for this valuable feedback. The size of the letters has been adjusted to be consistent in Figure 10A.

Comment 29: By gene expression, what is the bivariate correlation between GNPNAT1 and Slug?

Response: We thank the reviewer for raising this question. Based on gene expression, we observe a positive correlation between GNPNAT1 and Snai2 with Spearman's r = 0.22 (P = 8.2e-07)

Comment 30: Should you use SNAl2 instead of SLUG?

Response: Thanks. Slug is replaced by SNAI2 in this study.

We sincerely appreciate your thorough and thoughtful review, which has undoubtedly enhanced the quality of this manuscript. We hope that the revision we made adequately addressed your concerns. If you have any further suggestions, please do not hesitate to let us know.

Once again, thank you for your time and expertise in reviewing our manuscript.

Best regard,

YanYu

Harbin Medical University Cancer Hospital

Round 2

Reviewer 2 Report

Comments and Suggestions for Authors

The authors have significantly improved their manuscript.

Comments on the Quality of English Language

Minor English language editing is recommended.